

# On the forcings of the unusual QBO structure in February 2016

Haiyan Li[1,2,3], Robin Pilch Kedzierski[1], and Katja Matthes[1,4]

[1]Marine Meteorology Department, GEOMAR Helmholtz Centre for Ocean Research Kiel, Kiel, Germany
[2]School of Atmospheric Sciences, Sun Yat-Sen University, Zhuhai, China
[3]School of Electronic Information, Wuhan University, Wuhan, China
[4]Faculty of Mathematics and Natural Sciences, Christian-Albrechts-Universität zu Kiel, Germany

**Correspondence:** Haiyan Li (lihaiyan@whu.edu.cn)

**Abstract.** The westerly phase of the stratospheric Quasi-Biennial Oscillation (QBO) was reversed during Northern Hemisphere winter 2015/2016 for the first time since records began in 1953. Recent studies proposed that Rossby waves propagating from the extratropics played an important role during the reversal event in 2015/2016. Building upon these studies, we separated the extratropical Rossby waves into different wavenumbers and time-scales by analyzing the combined ERA-40 and ERA-Interim reanalysis zonal wind, meridional wind, vertical velocity and potential vorticity daily mean data from 1958 to 2017. We find that both synoptic and quasi-stationary Rossby waves are dominant contributors to the reversal event in 2015/2016 in the tropical lower stratosphere. By comparing the results for 2015/2016 with two additional events (1959/1960 and 2010/2011), we find that the largest differences in Rossby wave momentum fluxes are related to synoptic-scale Rossby waves of periods from 5-20 days. We demonstrate for the first time, that these enhanced synoptic Rossby waves at 40 hPa in the tropics in February 2016 originate from the extratropics as well as from local wave generation. The strong Rossby wave activity in 2016 in the tropics happened at a time with weak westerly zonal winds. This coincidence of anomalous factors did not happen in any of the previous events.

In addition to the anomalous behavior in the tropical lower stratosphere in 2015/16, we explored the forcing of the unusually long-lasting westerly zonal wind phase in the upper stratosphere (at 20 hPa). Our results reveal that mainly enhanced Kelvin wave activity contributed to this feature. This was in close relation with the strong El Niño event in 2015/2016, which forced more Kelvin waves in the equatorial troposphere. The easterly or very weak westerly zonal winds present around 30-70 hPa allowed these Kelvin waves to propagate vertically and deposit their momentum around 20 hPa, maintaining the westerlies there.

## 1 Introduction

The variability of zonal winds in the tropical lower stratosphere (100 to 10 hPa) is dominated by the Quasi-Biennial Oscillation (QBO) with descending easterly and westerly zonal wind regimes and a varying period from 22 to 36 months (Baldwin et al., 2001). The QBO was discovered by Reed et al. (1961) and Ebdon (1960) independently and its characteristics have been observed for many decades (Lindzen and Holton, 1968; Holton and Lindzen, 1972; Kawatani and Hamilton, 2013).



The tropical QBO not only influences the tropical stratosphere, but also impacts the tropical troposphere, for example the Hadley circulation (Gray et al., 1992) and the Madden-Julian oscillation (Yoo and Son, 2016). Furthermore, the QBO impacts the extratropical stratosphere by modulating the strength of the stratospheric polar vortex (Baldwin et al., 2001; Holton and Austin, 1991; Holton and Tan, 1980; Gray et al., 2004). The westerly mean flow in the tropical stratosphere generally favors the penetration of planetary waves into the tropical regions even across the equator without encountering a critical limit. The wave-mean flow interaction theory proposed that vertically propagating equatorial waves are the main forcing mechanism for the QBO through the selective filtering by the background wind (Lindzen and Holton, 1968; Holton and Lindzen, 1972). The vertically propagating equatorial waves deposit their momentum into the regions where the background zonal wind is equal to the wave phase speed. This mechanism explains the descent of the wind regimes alternating the QBO phase. The responsible equatorial waves include Kelvin waves, equatorial Rossby waves and smaller scale gravity waves (Lindzen and Holton, 1968; Holton and Lindzen, 1972).

Kelvin waves are excited by tropospheric convection and propagate upward and eastward (Lindzen and Holton, 1968; Holton and Lindzen, 1972; Baldwin et al., 2001). The eastward propagating Kelvin waves provide the main eastward acceleration for the initiation of the QBO westerly phase. In contrast, the westward propagating Rossby waves provide the main westward acceleration for the initiation of the QBO easterly phase. The period of the stratospheric QBO depends on the amount of vertically propagating equatorial waves (Baldwin et al., 2001).

In February 2016, the regular descent of the QBO westerly phase was interrupted and reversed near 40 hPa. The reversed westerly zonal mean zonal wind at 40 hPa (shown in Figure 1b) is named "reversal event" in the following. This reversal event occurred for the first time in 2016 since the QBO is recorded (Naujokat, 1986). Several studies tried to explore the possible reasons for the anomalous QBO behavior (Osprey et al., 2016; Dunkerton, 2016; Coy et al., 2017; Tweedy et al., 2017; Barton and McCormack, 2017; Lin et al., 2019). Their results showed that Rossby waves propagating from the extratropical Northern Hemisphere might have been the most likely cause of the reversed westerly zonal wind at 40 hPa. Coy et al. (2017) also found high amounts of momentum flux divergence present in the tropical lower stratosphere in 1987/1988 and 2010/2011, but the westerly zonal wind at the equator did not reverse as a consequence of this extratropical Rossby wave activity. Another possibility for the enhanced Rossby wave activity in the equatorial stratosphere is local wave generation from instability within the QBO westerlies. Coy et al. (2017) calculated the meridional gradient of the potential vorticity field ($\bar{q}_\phi$, (Andrews et al., 1987)) during the anomalous QBO reversal event. Regions with negative $\bar{q}_\phi$ indicate the presence of barotropic shear instability associated with the QBO winds. Coy et al. (2017) found that $\bar{q}_\phi$ was positive during the reversal event in 2015/2016, implying an unlikely instability of the large-scale flow. However, in the model study by Hitchcock et al. (2018) it is suggested that the $\bar{q}_\phi$ distribution present in February 2016 favored wave flux convergence over a narrow region, providing a dynamical feedback, not the source of the wave forcing. One goal of our study is to investigate the responsible waves for the reversal event in the lower stratosphere in February 2016 by separating the contributions of individual wavenumbers and different timescales.

As highlighted in previous studies, the stratospheric QBO can interact with other large scale oscillations. ENSO events influence tropical convection. The ENSO warm phase (El Niño) enhances the tropical tropospheric convection and the ENSO cold phase (La Niña) decreases the tropical tropospheric convection (Maury et al., 2013; Yang and Hoskins, 2013). Through





modulating the tropospheric convection, ENSO could affect the equatorial wave behavior and hence the forcing of the QBO (Maury et al., 2013; Yang and Hoskins, 2013; Schirber, 2015; Hansen et al., 2016; Christiansen et al., 2016). Based on this mechanism, many studies investigated the interaction between ENSO and the QBO. Taguchi (2010) and Yuan et al. (2014)

revealed that the QBO has a weaker amplitude and faster phase speed during El Niño conditions by investigating radiosonde data from 1953 through 2008. Recently, some studies (Barton and McCormack, 2017; Hirota et al., 2018) highlighted the potential role of the exceptionally strong El Niño conditions on the reversed QBO westerlies in 2015/2016. Their results agree with a modeling study by Calvo et al. (2010), which found that the subtropical zonal wind has a westerly tendency in the Northern Hemisphere in conjunction with a strong El Niño in winter (NDJ). Newman et al. (2016) reported that the QBO

westerlies propagated upward instead of the regular downward propagation above 30 hPa in 2015/2016. The upward migration of QBO westerlies above 30 hPa caused the westerly zonal winds to last unusually long at 20 hPa in 2015/2016. Another goal of our study is to explore the responsible waves and the role of the strong El Niño event for the upward migration and unusually long westerly zonal mean zonal wind regime at 20 hPa.

The remaining parts of this paper are organized as follows. In section 2, the data and methods used in our study are

described. The possible reasons for the reversed westerly zonal wind at 40 hPa are explored in section 3.1. The Rossby waves are divided into quasi-stationary Rossby waves and faster Rossby waves. Their contributions and sources (propagating from the extratropics or generated locally in the tropics) during the QBO westerly phase reversal event are investigated in subsections 3.1.1 and 3.1.2, respectively. We explore possible mechanisms for local Rossby wave generation during the reversal event in subsection 3.1.3. Section 3.2 will analyze the unusual behavior of Kelvin waves and the influence of the strong El Niño event

during the long-lasting westerly zonal mean zonal wind phase at 20 hPa. The summary and conclusions are given in section 4.

## 2 Data and Methods

For this study, we use the combined European Center for Medium-Range Weather Forecasts (ECMWF) ERA-40 and ERA-Interim reanalysis data sets (Uppala et al., 2005; Dee et al., 2011). We use daily averaged pressure level data from 1958 to 2017 including zonal wind, meridional wind, vertical velocity and potential vorticity with a horizontal resolution of 2.5° longitude

× 2.5° latitude. The ERA-40 and ERA-Interim reanalysis data sets have 23 and 37 pressure levels in the vertical direction, respectively, from 1000 hPa to 1 hPa. We use a merged dataset of ERA-40 and ERA-Interim on the 23 pressure levels they have in common (Blume et al., 2012). The NOAA Extended Reconstructed Sea Surface Temperature (SST) V4 was used to calculate the ENSO index. Furthermore, we also used the monthly Outgoing Long-wave Radiation (OLR) data to represent the amount of convective activity from https://www.esrl.noaa.gov/psd/data/gridded/data.interp_OLR.html.

## 95 2.1 Index Definitions

We define the QBO phase using the monthly zonal mean zonal wind at 40 hPa (averaged from 50 hPa to 30 hPa) and averaged over 2.5°S-2.5°N (U40) (Hansen et al., 2016). The westerly QBO phase (QBOW) is defined when U40 is greater than $2 m/s$ and the easterly QBO phase (QBOE) is defined when the U40 is less than -2 $m/s$ (shown in Figure 1b) following Hansen





et al. (2016). To examine the reliability of the QBO index, we compared the ERA-40 and ERA-Interim reanalysis data with

the monthly zonal mean zonal wind at 40 hPa from the Free University of Berlin (FUB) data set (FU40) (Naujokat, 1986). The amplitude of the QBO index is only slightly larger in FU40 compared with U40 (not shown). The index of FU40 is an average of three stations (Canton Island station (1953-1967), Gan/Maledive Islands station (1968-1975) and Singapore station (1976-present)), all of them at a similar latitude around the equator (Naujokat, 1986). Note that, the result of U40 comes from the zonal mean and averaged from 2.5°S to 2.5°N. The U40 and FU40 agree well in the duration and phase of zonal wind

regimes, suggesting that the result of U40 is robust and the difference between U40 and FU40 are negligible. We therefore use the ERA-reanalysis dataset for the rest of this study. Similar to U40, we also defined the direction of zonal wind at 20 hPa (U20). The westerly zonal wind was defined when U20 is greater than 2 $m/s$ and when U20 is less than -2 $m/s$ it is defined as easterly zonal wind (as shown in Figure 1a).

The ENSO index is obtained as deviations from the monthly climatology, with 5 months running average in the Niño 3.4

region (70°W-120°W, 5°S-5°N)(Hansen et al., 2016). An El Niño event is defined when the ENSO index is greater than 0.5 $K$ and a La Niña event is defined when the ENSO index is less than -0.5 $K$ (shown in Figure 1c).

### 2.2 Wave filtering

In order to explore the roles of tropical and extratropical waves during the unexpected QBO reversal event in 2015/2016, we extracted Rossby and Kelvin wave perturbations in the wavenumber-frequency domain by applying the "kf-filter" function (a

two-dimensional Fast Fourier Transform in longitude and time; Schreck (2009)) on horizontal winds and vertical velocity. Previous studies revealed that the reversal event near 40 hPa in 2015/2016 has been caused by Rossby waves propagating from the Northern extratropical regions (Osprey et al., 2016; Newman et al., 2016; Coy et al., 2017). We extracted the equatorial Rossby waves following their dispersion curve (Wheeler and Kiladis, 1999). It is difficult to define a single filter in the wavenumber-frequency domain to extract extratropical and tropical Rossby waves that will work for the whole time period, all latitudes and

different stratospheric levels. The propagation of waves is affected by the background winds through Doppler shifting. Both tropical and extratropical waves can travel in the same wavenumber-frequency domain depending on the background zonal winds. In order to extract the Rossby waves from 60°N to 60°S we used wide boxes in the wavenumber-frequency domain (see Table 1) instead of defining certain dispersion curves following (Pilch Kedzierski et al., 2017). It is difficult to separate the tropical and extratropical Rossby waves with this filtering technique. We can track their origins by dividing them into different

wavenumbers and time-scales, with the use of time-height and time-latitude sections of each Rossby wave type. The detailed information about the Rossby waves filter boundaries is shown in Table 1. The filter boundaries for Rossby waves are -0.5 and -3.5: negative wavenumbers indicate westward-propagating waves. In our study we focus on planetary-scale westward propagating Rossby waves with wavenumbers 1 to 3 and periods from 5 to 70 days. Extratropical Rossby waves can also propagate eastward relative to the ground with strong westerly mean zonal winds, but their activity and fluxes are limited to the

winter stratospheric polar vortex, therefore not affecting tropical regions. We confirm that momentum fluxes of relevance for tropical regions only originate from westward propagating waves (see and compare Figures 3 and S1 from the supplement), and thus the eastward propagating Rossby waves are not included in our study. The westward propagating Rossby waves pro-





duce a westward acceleration of the background zonal wind and hence weaken the westerly zonal wind in the extratropics during Northern Hemisphere (NH) winter. First, the total amount of Rossby wave activity (periods of 5-70 days) is presented

135 in section 3.1. Then, we focus on the two dominant time-scales: quasi-stationary (20-40 days) and faster (5-20 days) Rossby waves in subsections 3.1.1 and 3.1.2, respectively. In order to explore their sources and contributions during the 2015/2016 reversal event, for each time-scale we further separate the filtered Rossby waves into individual wavenumbers 1, 2, and 3. We also study the Kelvin wave activity (wavenumbers 1 to 14, periods of 4-30 days, see table 1) in section 3.2.

 Please note that the output of the filter near the temporal ends (years 1958 and 2017) of the data set in our analysis is

140 neglected because the amplitude of the filtered waves is underestimated there. The monthly climatology is calculated from January 1959 through December 2014, to avoid any influence of the unique structure of the stratospheric QBO in 2015/2016.

## 2.3 Momentum Flux Calculation

The contribution of each wave type defined in section 2.2 during the 2015/2016 QBO reversal event is estimated by calculating the horizontal ($Mf_h$) and vertical ($Mf_v$) momentum fluxes:

145 $$Mf_h = \overline{u'v'} \tag{1}$$

$$Mf_v = \overline{u'w'} \tag{2}$$

 $u'$, $v'$ and $w'$ denote the perturbations (waves) in the zonal wind, meridional wind and vertical velocity. The overbar denotes the zonal mean.

 In the following, only results for vertical momentum fluxes for Kelvin waves will be presented, since no relevant contribution

150 from Rossby waves was detected (see and compare Figures 12a and S2 from the supplement.)

## 2.4 Barotropic and Baroclinic Instability Calculation

We explore a possible generation mechanism for enhanced Rossby wave activity in section 3.1.3. Shuckburgh et al. (2001) revealed that barotropic and baroclinic instability in tropical regions could be associated with QBO westerlies. To investigate barotropic and baroclinic instability, the meridional gradient of potential vorticity in 2015/2016 was calculated from 70 hPa to

155 10 hPa following Andrews et al. (1987) as used in Shuckburgh et al. (2001) and Coy et al. (2017).

$$q_\phi = 2\Omega \cos\phi - [\frac{(\bar{u}\cos\phi)_\phi}{a\cos\phi}]_\phi - \frac{a}{\rho_0}(\frac{\rho_0 f^2}{N^2}\bar{u}_z)_z \tag{3}$$

Where $\Omega$ is the Earth's rotation frequency, $a$ is the Earth's radius, $\bar{u}$ denote the zonal and time averaged zonal wind, $\rho_0$ is the basic state density, $z$ is the log pressure vertical coordinate, $\phi$ is the latitude and $N^2$ denotes the Brunt-Väisälä frequency squared. A negative meridional potential vorticity gradient is indicative of barotropic-baroclinic instability and hence is used

160 as condition for local Rossby wave generation in section 3.1.3.



## 3 Details and Mechanisms of the Unusual QBO Behavior in 2015/2016

The result chapter is divided into two parts. First, the interruption of the westerly zonal wind in the lower stratosphere (40 hPa) in February 2016 and possible reasons are investigated (section 3.1). Then, the reasons for the unusually long lasting westerly zonal winds in the upper stratosphere (20 hPa) are investigated in more detail. Figure 1b shows the reversal of the

westerly zonal wind regime and the onset of easterlies around 40 hPa in February 2016. A weakening of westerly zonal winds occurred in several earlier winters, for example, in 1959/1960 and 2010/2011 winter (Coy et al., 2017), but so far no other winter resulted in a reversal of the wind. This is exceptional to the winter 2015/2016 and resulted in a number of publications. We will add significant new aspects to the existing literature. As shown in Figure 1a, the westerly zonal wind around 20 hPa lasted unusually long in 2015/2016. Newman et al. (2016) and Kumar et al. (2018) noticed the unusual behavior of westerly

zonal wind above 30 hPa but none of them explored it in detail. The goal of this study is to investigate both, the unusual behavior of the QBO in winter 2015/2016 in the lower and upper stratosphere in more detail than earlier studies.

### 3.1 Interruption of the westerly zonal wind at 40 hPa

Figure 1b shows that the westerly zonal wind was weaker in a number years (e.g., 1959/1960, 2010/2011) but the westerly zonal wind only reversed its direction in the 2015/2016 winter. Previous studies (Osprey et al., 2016; Newman et al., 2016; Coy

et al., 2017) reported that the reversal event in 2015/2016 winter at 40 hPa was caused by enhanced Rossby wave activity in the tropical stratosphere, mainly of extratropical origin. Based on these studies, we investigate the Rossby wave activity in the tropical stratosphere during the reversal event in more detail by looking not only at wavenumbers 1-3 but also by separating quasi-stationary Rossby waves (20-40 days) and faster Rossby waves (5-20 days).

First, to show the total amount of Rossby wave activity, we extracted all Rossby waves with periods of 5-70 days and with

wavenumbers from 1 to 3. Figure 2 illustrates the time-height cross section of the squared Rossby wave anomalies in zonal wind averaged over the tropics (10°S-10°N) with respect to the monthly climatology (1959-2014). As shown in Figure 2, the Rossby wave activity was enhanced in the tropics before the westerly zonal wind reversed near 40 hPa in February 2016. Enhanced Rossby wave activity in the tropics usually occurs in November and December but seldom in January and February in the NH (O'Sullivan, 1997). However, we cannot find enhanced Rossby wave behavior at lower or higher levels in the tropics,

which suggests that it did not originate from vertical propagation. Besides vertical propagation, Rossby waves can propagate in the meridional direction (O'Sullivan, 1997; Osprey et al., 2016). We choose two additional cases to compare the results with the 2015/2016 case: 1959/1960 and 2010/2011. Both have increased horizontal Rossby wave momentum fluxes (shown in Figure 1d). We investigated the role of meridionally propagating Rossby waves during the reversal event as previous studies (Osprey et al., 2016; Newman et al., 2016; Coy et al., 2017; Tweedy et al., 2017).

Figure 3 shows the time-latitude cross section of the horizontal Rossby waves momentum flux during three related cases (marked as blue and red vertical lines in Figure 1b). The result is in agreement with the findings of Barton and McCormack (2017) which found that westerly subtropical zonal winds favor extratropical Rossby waves to propagate into the tropics. Rossby wave horizontal momentum fluxes maximize in the extratropics and the magnitude generally decreases with decreasing





latitude. Equatorward propagation of Rossby waves is observed in all three example winters (Figure 3) in agreement with
Osprey et al. (2016) and Coy et al. (2017). However, the specific time-scales and wavenumbers of the responsible Rossby
waves have not been explored before. While previous studies discussed the extratropical Rossby wave origin, a tropical origin
of the enhanced Rossby waves in February 2016 is still unclear and will be explored in more detail in section 3.1.3.

Our filtered Rossby waves in the tropics include Rossby waves generated in the tropics and in the extratropics. In order to
detect where the Rossby waves were generated, we analyzed the time-series of the mean Rossby wave horizontal momentum
flux separately in the tropics (10°S-10°N) and extratropics (35°N-45°N), for the three selected winters 1959/1960, 2010/2011
and 2015/2016 (Figure 4).

Figure 4 shows that the horizontal Rossby wave momentum fluxes peak as expected in the extratropics but some also peak
in the tropics. For example, the extratropical momentum fluxes in February 2015 and 2010 do not coincide with any wave
activity in the tropics. The horizontal Rossby wave momentum flux were not able to propagate into the tropics due to the
prevailing easterly background zonal wind in the subtropics around 20°N (see Figure 3). Figure 4 suggests that the peaks of
the horizontal Rossby wave momentum flux in the tropics in January 1960, November 2010 and February 2016 are related to
extratropical Rossby waves (highlighted with red triangles in Figure 4), since there is a few days lag between the peaks in the
tropics and extratropics. This implies that they need some days to propagate from the extratropics into the tropics, which can
be easily seen in Figure 3.

We now focus on faster Rossby waves (5-20 days, w0520) and quasi-stationary Rossby waves (20-40 days, w2040) as
described in section 2.2. We calculated the mean horizontal momentum flux for the total as well as the contribution from fast
and quasi-stationary Rossby waves in the tropics (Figure 5). Figure 5 shows that the maximum amplitude of the horizontal
Rossby wave momentum fluxes occurred in February 2016 (3.43 $m^2/s^2$). In addition, the Rossby w0520 horizontal momentum
flux was stronger during January 1960, November 2010 and February 2016 than Rossby w2040, which had similar values in all
three cases. This suggests a dominant contribution from faster Rossby waves (w0520) during the QBO reversal event, which
was not noted in previous studies (Osprey et al., 2016; Newman et al., 2016; Coy et al., 2017).

From Figure 5 we conclude that quasi-stationary Rossby waves contribute significantly to the observed momentum fluxes in
the tropics, in a fairly similar way in all three cases. Faster Rossby waves (w0520) are responsible for most of the case to case
variability, being especially dominant in February 2016. It is worth analyzing the contributions from individual wavenumbers
for Rossby waves of both time-scales, which also enables tracing their origins more precisely. Such a detailed analysis of
the wave forcings of the QBO reversal event in February 2016 has not been performed yet and may lead to results of high
interest. Lin et al. (2019) reported an important contribution to the 2016 QBO disruption made by a single wave packet,
with dominating wavenumbers 1-3 in its wave spectrum. Their study was performed at a similar vertical level as ours (35.8
hPa and 40 hPa respectively), and our analysis will describe the evolution of its different components in further detail. The
following subsections 3.1.1 and 3.1.2. will focus on behavior of quasi-stationary and faster Rossby waves, respectively,
whereas subsection 3.1.3 will explore possible mechanisms for an observed locally generated wave whose contribution was
important and only present in February 2016.





### 3.1.1 The Role of Quasi-Stationary Rossby waves

Firstly, we discuss the contribution of quasi-stationary Rossby waves (Rossby w2040, light blue lines in Figure 5). As shown
in Figure 5, the Rossby waves w2040 were enhanced in January 1960, November 2010 and February 2016 (red triangles in
Figure 5). The Rossby waves w2040 horizontal momentum flux had maximum values (about 0.78 $m^2/s^2$) in February 2016
in the tropics at 40 hPa. In November 2010 and January 1960, the Rossby waves w2040 horizontal momentum flux were
0.41 $m^2/s^2$ and 0.66 $m^2/s^2$, respectively, close to the values in February 2016. We calculated the horizontal momentum flux
from Rossby waves w2040 with wavenumber 1 and 2, respectively. Then we analyzed their variation in time-latitude sections
(shown in Figures 6 and 7). Figures 6 and 7 show that the contribution of Rossby waves w2040 with wavenumber 1 and 2
was weaker but similar in November 2010. While in January 1960 and February 2016 were characterized by a strong Rossby
wave w2040 with wavenumber 1. However, the total contribution of quasi-stationary Rossby waves was similar in all cases as
shown in Figure 5. In order to highlight the latitudinal variation of the Rossby waves w2040 with wavenumber 1, we analyzed
its horizontal momentum flux at the equator, 10°N, 20°N and 30°N during each case, as shown in Figure 8, which reveals that
the peaks of Rossby waves w2040 (with wavenumber 1) in the extratropics occurred earlier at higher latitudes. This means that
the Rossby waves w2040 (with wavenumber 1) probably originates from the extratropics in January 1960, November 2010 and
February 2016 at 40 hPa. Figures 5, 6 and 7 reveal that quasi-stationary Rossby waves have an important contribution to the
enhanced Rossby wave activity in the tropics, mainly by wavenumber 1 (and 2 to a lesser degree) of extratropical origin.

### 3.1.2 The Role of Faster Rossby Waves

Lin et al. (2019) reported that the low frequency waves (slower than 0.15 cpd, or 6-7 day frequency) with wavenumbers 1-3
were important contributors to the reversal event in February 2016. This category would include the faster and quasi-stationary
Rossby waves from our analysis. In the previous sections we demonstrated that the case to case differences are mostly due to
the faster wave type (5-20 day frequencies), which will be explored in this section.

We explored the activity of faster Rossby waves (Rossby w0520, dark blue lines in Figure 5), separating wavenumbers 1, 2,
and 3 which will be referred to as W1, W2 and W3 in the following. The time-latitude structure of the horizontal Rossby waves
w0520 momentum flux for each case is shown in Figure 9. The contribution of Rossby waves w0520 is largest in February
2016. Figures 9b and 9c show that the amplitude of the horizontal Rossby waves w0520 momentum flux was stronger in the
extratropics in January 1960 and November 2010. In February 2016, there were two peaks of Rossby waves w0520 horizontal
momentum flux, in the extratropics (around 40°N) and in the tropics (around 15°N) as shown in Figure 9a. We therefore
focus now on the contributions of W1, W2 and W3 separately during the 2016 reversal event. We analyzed the time-latitude
cross section of their horizontal momentum flux from November 2015 to April 2016 at 40 hPa (as shown in Figure 10). The
result reveals that in particular W2 and W3 activity was enhanced in 2015/2016 but the activity due to W1 was very weak.
The maximum horizontal momentum flux of W2 occurred around 40°N in early February 2016 (Figure 10c), and propagated
equatorward since early February 2016. Although it is difficult to distinguish in Figure 10, W2 activity shows some days lag
between 40°N and 15°N (see Figure S4), indicating its extratropical origin and equatorward propagation.



The peak of W3 occurs in early February 2016, around 15°N (as shown in Figure 10d). Furthermore, as shown in Figure 2, the enhanced Rossby wave activity was concentrated on the lower stratosphere (from 50 hPa to 30 hPa). Figures 2 and 10d suggest that the W3 peak does not originate from the extratropics or the vertical propagation, therefore the remaining possibility is local wave generation. The record high horizontal momentum flux from Rossby waves in February 2016 at 40 hPa seems to be the combination of a quasi-stationary W1 (Rossby waves w2040), a faster W2 generated in the extratropics, as well as a possibly local generated W3. Figure 10e reveals that the peaks of W1 (Rossby waves w2040) and W2 occurred earlier compared to the peak of the W3 in early February 2016. This result implies that not all components of the wave packet described by Lin et al. (2019) were of extratropical origin. Strictly speaking, all wave components of a wave packet should be traveling together, while our analysis shows that for W3 this is not the case. The amplitudes of W1 (Rossby waves w2040), W2 and W3 are 0.74 $m^2/s^2$, 0.931 $m^2/s^2$ and 1.312 $m^2/s^2$, and hence W3 was stronger than W2 and W1 (Rossby waves w2040). The locally generated W3 only occurred in February 2016 and we do not find such behavior in other cases, whose Rossby wave contributions are exclusively of extratropical origin (see Figures S5 and S6 for comparison with Figure 10).

### 3.1.3 The Possible Source of Local Rossby Wave Generation

Figure 10e demonstrates that the quasi-stationary W1 (Rossby waves w2040) and the faster W2 and W3 have stronger horizontal momentum fluxes in February 2016 at 40 hPa around 15°N. This suggests that the possibly locally generated W3 was as important as the Rossby waves which propagated from the extratropics during the reversal event in 2015/2016. Therefore it is important to investigate the possible source of the locally generated Rossby waves. As demonstrated in the previous section, the W3 generated in early February 2016 is not related to vertical or meridional propagation. In this section we will discuss two possible generation mechanisms: barotropic and baroclinic instability and nonlinear interactions.

We analyzed the barotropic and baroclinic instability in the lower stratosphere with the meridional gradient of potential vorticity at 40 hPa in February 2016 similar to Coy et al. (2017). The meridional gradient of potential vorticity was calculated at 70 hPa, 50 hPa, 40 hPa, 30 hPa, 20 hPa and 10 hPa from January 2015 to December 2016 (as shown in Figure S7). Figure S7 reveals that the meridional gradient of potential vorticity was greater than zero in February 2016 at 40 hPa and below, meaning that the atmosphere was stable during our analysis period below 40 hPa. The meridional gradient starts to be negative which indicates barotropic and baroclinic instability (see also section 2.4) in late March 2016. This indicates that W3 maximizing in February 2016, so one month before, was not generated by instability. Our result is consistent with Coy et al. (2017) which showed that $\bar{q}_\phi$ was positive during the reversal event in 2015/2016, implying that instability of the large-scale flow was unlikely. Hitchcock et al. (2018) also suggested that the distribution of the $\bar{q}_\phi$ could have directed and concentrated wave fluxes to this narrow region.

Another possible mechanism for the local generation of W3 is the nonlinear interaction between several different Rossby waves. Previous studies (Reznik et al., 1993; Huang et al., 2009; Tamarin et al., 2015) pointed out that nonlinear coupling processes could occur between different waves. Based on the wave interaction theory (Reznik et al., 1993; Huang et al., 2009),





two waves could force a third wave if the first two waves satisfy the resonant interaction condition:

$$\omega(k_1) \pm \omega(k_2) = \omega(k_3), k_1 \pm k_2 = k_3 \tag{4}$$

Where $\omega$ and $k$ are the frequency and wavenumber of the Rossby waves.

We analyzed the Fourier spectrum of the Rossby wave perturbations in zonal wind with wavenumbers 1, 2 and 3 at 40 hPa averaged over 10°N-20°N from January 2016 to February 2016 (Figure 11). Figure 11 shows that W1 had the strongest power at a frequency of 0.033 day$^{-1}$. W2 has two strong peaks, at frequencies 0.033 day$^{-1}$ and 0.066 day$^{-1}$. W3 has the most complex peaks, the stronger peaks are corresponding to the frequencies are at 0.054 day$^{-1}$ and 0.066 day$^{-1}$. Equation 4

reveals that the W1 with frequency of 0.033 day$^{-1}$, W2 with frequency of 0.033 day$^{-1}$ and W3 with frequency 0.066 day$^{-1}$ almost perfectly satisfy the nonlinear resonance interaction condition (see equation 4). The W3 power peak at 0.066 day$^{-1}$ frequency belongs to the filtered Rossby waves w0520 whose horizontal momentum fluxes are displayed in Figures 9 and 10. Moreover, the quasi-stationary Rossby wave W1 and the faster Rossby wave W2 occur first and the faster Rossby wave W3 appears to peak afterwards (as shown in Figure 10e). These results imply that nonlinear interactions and resonance between

the quasi-stationary W1 and the faster W2 which came from the extratropics, generated W3 locally at 15°N, 40 hPa.

Both W2 and W3 had important contributions to the enhanced Rossby wave horizontal momentum flux in the tropics (Figure 10). Our results from Figures 10 and 11 indicate that the mechanism for the local generation of W3 at 15°N, 40 hPa was nonlinear interaction and resonance between a quasi-stationary Rossby wave W1 and a faster Rossby wave W2 of extratropical origin. The reasons for the special behavior and the exact timing and location of the waves would need further work and are

beyond the scope of this paper. Coy et al. (2017) noted that the horizontal momentum fluxes were very concentrated at a specific height range and latitude during the QBO reversal of the 2015/2016 winter, which is in agreement with our result. The very rare origin of W3 at 15°N and its important contribution to these momentum fluxes could explain why it was so localized, since the conditions for nonlinear interaction and resonance must have been very localized too. In addition, W3 cannot be an equatorial wave mode in principle, otherwise its amplitude would be by definition maximizing at the equator.

In summary, we explored the Rossby wave horizontal momentum flux in the extratropics as well as in the tropics for three different cases (1959/1960, 2010/2011 and 2015/2016). The results show that the Rossby wave horizontal momentum flux in the tropics is dominated by a quasi-stationary Rossby wave W1 originating from the extratropics, as well as enhanced activity of faster Rossby waves which amount for the largest case-to-case differences (e.g., an equatorward propagating extratropical Rossby wave W2 and the locally generated Rossby wave W3 in the 2015/2016 case). During the 2015/2016 reversal event at

40 hPa, the locally generated Rossby wave W3 had the largest and most localized contribution (among faster Rossby waves) to the record horizontal momentum fluxes in the tropics.

Whereas the total structure from W1, W2 and W3 around 15°N resembles that of a wave packet as described by Lin et al. (2019), our results suggest that only the W1 and W2 components are of extratropical origin, both with similar frequency. The W3 component seems to originate later from the resonant interaction between W1 and W2.

The peaks of the Rossby wave horizontal momentum flux almost correspond to the strongest westerly zonal wind (greater than 10 $m/s$) in November 2010 (see Figure 5). Meanwhile, in January 1960 and February 2016 the peaks of the Rossby





wave horizontal momentum flux correspond to a weaker zonal wind (less than 5 $m/s$). Through a comparison of the results in January 1960 and February 2016 when the background zonal winds were similar (see Figures 9a and 9c), we found that the enhanced Rossby wave activity is an important factor for the reversal event in 2015/2016. The reversal event not only depends

on the enhanced Rossby wave activity in the tropics but also on the different background zonal winds. The stronger Rossby wave horizontal momentum flux together with the weaker westerly zonal mean zonal wind both played significant roles for the reversed westerly zonal wind at 40 hPa in February 2016. Lin et al. (2019) pointed out that the westerly flow was decelerated by mixed Rossby-gravity wave momentum fluxes (MRG, 0.15-0.5 cpd frequency, ~6-2 day periods) prior to the appearance of the low-frequency waves, i.e. the W1-3 wave packet.

## 3.2   Reasons for the Unusually Long Westerly Zonal Wind at 20 hPa

This subsection will focus on the behavior of Kelvin waves during the unusually long westerly zonal wind at 20 hPa in 2015/2016. The possible role of the strong El Niño event during the long lasting westerly zonal wind will be also explored. As shown in Figure 1a the westerly zonal mean zonal wind lasted unusually long in 2015/2016 at 20 hPa. Previous research mainly focused on the interruption of the westerly zonal wind near 40 hPa in 2015/2016, and only a few other studies (Newman

et al., 2016; Coy et al., 2017; Kumar et al., 2018) noticed the unusual behavior of the zonal mean zonal winds above 30 hPa. By performing a statistical analysis of the period of the westerly zonal mean zonal wind at 20 hPa, we found that the average period of westerly zonal mean zonal wind at 20 hPa is 10.6 months between 1958 to 2014 (Figure 12). In 2015/2016 the westerly zonal wind lasted 23 months (from May 2015 to March 2017) which is much longer than the maximum period (16 months) from 1958 to 2014. No similar westerly zonal wind period has ever been observed.

Newman et al. (2016) noticed that the westerly zonal wind started to migrate upward instead of downward from November 2015 to April 2016 above 30 hPa. The extended westerly zonal wind and the upward propagating westerly zonal wind above 30 hPa could result from wave-mean flow interaction. Since Kelvin waves are generated in the troposphere and propagate upward to the lower stratosphere where they initiate an eastward acceleration of the zonal mean zonal wind (Baldwin et al., 2001), they might play a role in the extended westerly zonal winds.

Figure 13 shows that the squared perturbations in zonal wind for Kelvin waves were enhanced from December 2015 to April 2016 from 50 hPa to 20 hPa. Figure 13 shows that the squared zonal wind perturbations had maximum amplitudes around the tropopause. Since the vertical momentum flux from Rossby waves (Figure S2) is very weak during this time, we will focus only on vertical momentum fluxes from Kelvin waves in the following (Figure 14).

Figure 14a demonstrates that the vertical momentum flux from Kelvin waves was strong during most of the easterly zonal

wind condition from the troposphere to the stratosphere in 2015. Furthermore, the vertical momentum flux from Kelvin waves was enhanced from December 2015 to April 2016 from 50 hPa to 20 hPa which agrees with the increased squared zonal wind perturbations of Kelvin waves (Figure 13). It should be noted that from December 2015 to around April 2016 near 20 hPa the background zonal mean zonal wind is westerly. Also in the levels down to 300 hPa, from December 2015 until February 2016, the mean zonal winds are weak westerlies or around zero. Kelvin waves were propagating upward at the time, while in





theory they only do within easterlies. One explanation for these seemingly contradicting facts is the possibility that a zonally
constrained - but large enough area of easterlies existed at the time, allowing synoptic-scale Kelvin waves to propagate upward.

We analyzed the vertical momentum flux from Kelvin waves during the westerly zonal wind conditions at 20 hPa from
1958 to 2016 (Figure 14b). The result shows that the maximum vertical momentum flux from Kelvin waves corresponds to
the beginning of the westerly zonal wind phase in general (Figure 14b). The vertical Kelvin wave momentum flux becomes

weaker when the westerly zonal wind gets stronger. The westerly zonal wind starts in May 2015 (around 3 $m/s$). Meanwhile
the vertical Kelvin wave momentum flux was around $0.4 \times 10^{-3} m^2 s^{-2}$. From late 2015 to early 2016 the amplitude of the
background zonal wind was similar to the background zonal wind in May 2015, but the vertical momentum flux from Kelvin
waves was much stronger. The amplitudes of the vertical Kelvin wave momentum flux in December 2015, January, February,
March and April 2016 are 0.88, 1.54, 1.31, 0.872 and $1.025 \times 10^{-3} m^2 s^{-2}$, respectively. They are much larger than the average

values ($0.45 \times 10^{-3} m^2 s^{-2}$) in westerly zonal wind conditions from 1958 to 2014. The vertical momentum flux from Kelvin
waves started to increase from December 2015 and ended in April 2016 which corresponds to the upward propagating westerly
zonal winds (Figure 2) reported in Newman et al. (2016). The unusually large vertical momentum flux from Kelvin waves
from December 2015 to April 2016 was deposited to the westerly zonal wind and resulted in additional eastward momentum
and hence a prolongation of the westerly winds at 20 hPa.

We also analyzed the temporal variation of the ENSO index and the convective activity in the tropics (Figure S8). The
ENSO index was larger than 2 $K$ (strong El Niño event) from October 2015 to January 2016. Tropospheric convective activity
is represented by the Outgoing Long-wave Radiation (OLR). The deviations of zonal mean OLR from the monthly climatology
had minimum values in February 2016 which corresponds to enhanced convective activity and at the same time to the enhanced
Kelvin wave activity at 20 hPa (Figure S8). Moreover, the convective activity and the vertical momentum flux from Kelvin

waves (at 20 hPa) have much stronger values in January, February, March and April 2016 which corresponds to the reversal
event at 40 hPa (Figure 14). Our results suggest that there is a close relationship between the strong El Niño event and enhanced
Kelvin wave activity in 2015/2016. We also calculated the squared perturbations in zonal wind for Kelvin waves as deviation
from the monthly climatology at 200 hPa during La Niña, Neutral and El Niño conditions (Figure 15). This indicates that
Kelvin wave activity is suppressed during La Niña and enhanced during El Niño conditions. The enhanced vertical Kelvin

wave momentum flux could be excited by enhanced convective activity in the troposphere which might be caused by the strong
El Niño event. The wind structure below 20 hPa that combined weak westerly and easterly zonal winds favored Kelvin waves
to propagate vertically. Moreover, as shown in Figure 14a, the westerly zonal wind was very weak in January and February
2016. In March and April 2016 the zonal wind was in its easterly phase below 30 hPa and weak westerly around 20 hPa. This
condition allowed the Kelvin waves to reach 20 hPa and deposit their vertical momentum to the background wind.

From our investigation, the extended westerly zonal wind near 20 hPa was possibly caused by enhanced Kelvin wave activity
in 2015/2016. The strong vertical momentum flux by Kelvin waves has a close relationship with the strong El Niño event in
2015/2016. The strong Kelvin waves in the troposphere together with the weaker zonal winds below 20 hPa finally lead to the
extended westerly zonal winds at 20 hPa. To our knowledge, the enhanced Kelvin wave activity and its relation to ENSO has
not been noted previously.





## 4 Conclusions


The QBO westerly phase was reversed by an unexpected easterly jet near 40 hPa and the westerly zonal wind lasted unusually long at 20 hPa during NH winter 2015/2016. By analyzing the horizontal momentum flux from Rossby waves in the extratropics and tropics we find that Rossby waves propagating from the northern extratropics were important contributors to the easterly jet around 40 hPa, in agreement with previous studies (Osprey et al., 2016; Newman et al., 2016; Coy et al., 2017). We additionally

explored different time-scales and wavenumbers of the Rossby waves and concluded that quasi-stationary Rossby W1 and faster Rossby W2 waves which propagated from the extratropics had important contributions to the reversed westerly zonal wind at 40 hPa in 2015/2016 (Figures 6 and 10). Furthermore, we find that during the reversal event a locally generated Rossby wave W3 around 15°N with a period of 5-20 days also played an important role (Figure 10). We explored two possible generation mechanisms for this locally generated Rossby wave: barotropic-baroclinic instability and nonlinear wave-wave interaction.

The locally generated Rossby wave w0520 with W3 in the tropics was unlikely caused by large scale instability (Figures 2 and 3). Our results imply that nonlinear interactions between the quasi-stationary Rossby wave W1 and faster Rossby wave W2 of extratropical origin, generated the local Rossby wave W3 around 15°N (Figure 11). The reasons for the special behavior and the exact timing and location of the waves is beyond the scope of this paper. In addition to the extratropical horizontal Rossby wave momentum fluxes and the locally generated Rossby waves in the tropics, the weaker background winds as compared to

two similar winters in 1959/1960 and 2010/2011 helped to favor this unusual reversal event in February 2016. Three important and novel take-home messages result from section 3.1 of our study:

1) The largest differences in horizontal Rossby wave momentum fluxes among all three investigated cases are related to faster Rossby waves (periods of 5-20 days) of extratropical origin.

2) The February 2016 case is unique, in the sense that it shows a local generation (in latitude and height) of a Rossby wave

W3, which is not observed in any other case and has the largest and most localized contribution to the observed horizontal momentum fluxes.

3) We relate local generation of the aforementioned W3 to a resonant interaction between quasi-stationary Rossby wave W1 and a faster Rossby wave W2 of extratropical origin that were present at 15°N.

Lin et al. (2019) reported that the reversal event of the QBOW in February 2016 had a dominant contribution from a wave

packet at 35.8 hPa. They also proposed that MRG played an important role for preconditioning this event. Hitchcock et al. (2018) revealed that the influence of nonlinear feedbacks of the $\bar{q}_\phi$ structure is important for the disruption of the QBOW in February 2016. Our results, studying the components of the above mentioned wave packets in more detail, are consistent with previous studies while highlighting the responsible mechanisms (local W3 generation from resonant interaction).

Besides the upward propagation of westerly zonal wind regime at 20 hPa in the 2015/2016 winter (Newman et al., 2016), we

find that the westerly zonal wind lasted unusually long (Figures 12 and 13). Our results suggest that the upward propagating and prolonged westerly zonal winds at 20 hPa could be caused by enhanced Kelvin wave activity (Figures 13 and 14). The enhanced Kelvin wave activity could be related to strong convective activity. Indeed, the strong El Niño event in 2015/2016 favored strong convective activity, which in turn excited stronger Kelvin wave activity. Under La Niña conditions Kelvin





wave activity is suppressed (Figure 15). The weaker zonal winds in the troposphere and lower stratosphere (related to the combination of anomalous factors around 40 hPa as summarized above) favored the upward propagation of Kelvin waves. The increased Kelvin wave activity then produced more eastward and upward acceleration of the zonal mean zonal winds and lead to the unusual westerly zonal wind structure at 20 hPa in 2015/2016.

*Acknowledgements.* We thank the European Center for Medium-Range Weather Forecasts (ECMWF) for the freely available ERA-40 and ERA-Interim data, and the Free University of Berlin for the provision of the QBO data. We thank NOAA for the freely available Sea Surface Temperature data. This work was carried out during a stay of Haiyan Li as a joint training Ph.D student with funding provided from the Chinese Scholarship Council at GEOMAR in Kiel (Germany). We thank Dr. Sebastian Wahl for his help with processing the ECMWF, OLR and SST data, and Dr. Wenjuan Huo for help with the figures.The article processing charges for this publication are covered by GEOMAR as a Research Centre of the Helmholtz Association.





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





**Table 1.** Exact boundaries of the Rossby wave and Kelvin wave filters.

|  | Period (days) | Wavenumber | Equivalent depth (m) |
|---|---|---|---|
| Rossby wave | 5∼70 | -3.5∼-0.5 | / |
| Rossby w0520 | 5∼20 | -3.5∼-0.5 | / |
| Rossby w2040 | 20∼40 | -3.5∼-0.5 | / |
| Kelvin wave | 4∼30 | 1∼14 | 6∼600 |





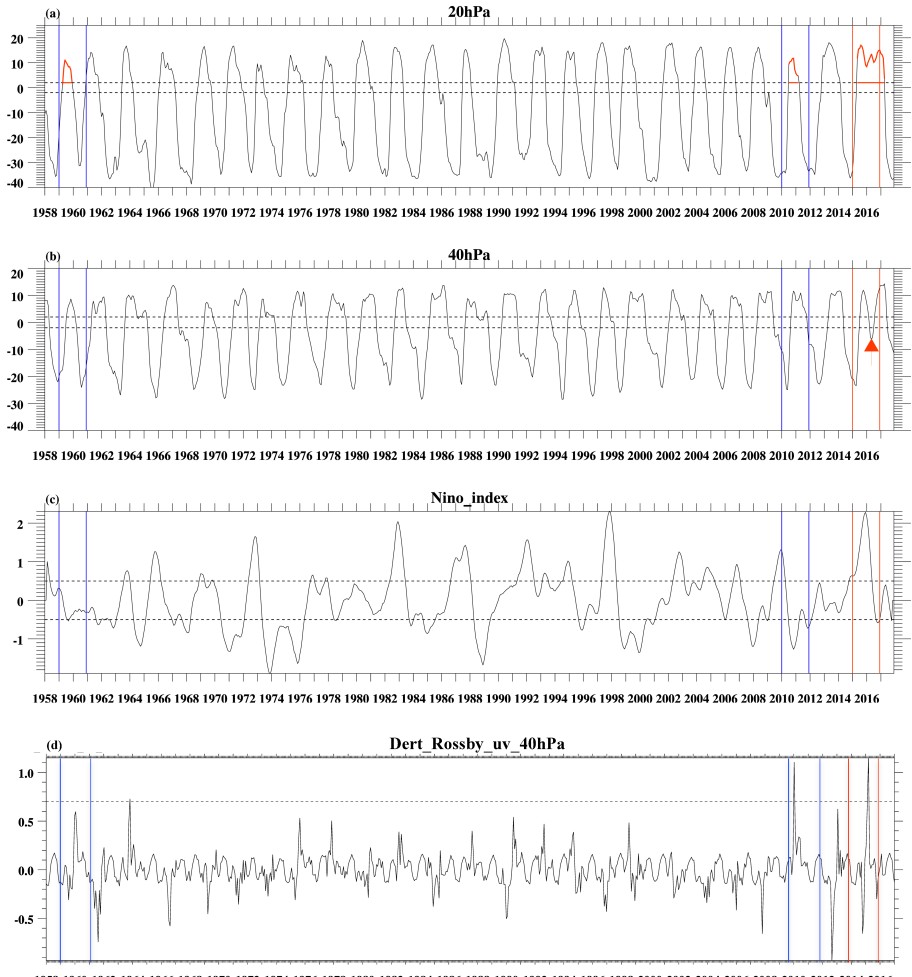

**Figure 1.** Zonal mean zonal wind at (a) 20 hPa, (b) 40 hPa, (c) ENSO index in the Niño 3.4 region. The vertical blue lines denote the time periods of 1959/1960 and 2010/2011 and (d) time series of the horizontal Rossby wave momentum flux with respect to the monthly climatology averaged over the tropics (from 10°S to 10°N) at 40 hPa from 1958 to 2017 (unit is $m^2/s^2$). The horizontal dashed lines denote the speed of 2 m/s and -2 m/s in panels (a) and (b). In panel (c) the horizontal dashed lines denote 0.5 K and -0.5 K, respectively. The horizontal dashed line denotes the value of 0.7 $m^2/s^2$ in panel (d). The vertical red lines denote the time period of 2015/2016 and the vertical blue lines denote the periods of 1959/1960 and 2010/2011. In panel (a) the red triangles and horizontal lines denote the westerly zonal wind and the westerly phase period during 1959/1960, 2010/2011 and 2015/2016 at 20 hPa. In panel (b) the red triangle denotes the reversed westerly zonal wind in 2015/2016 at 40 hPa.

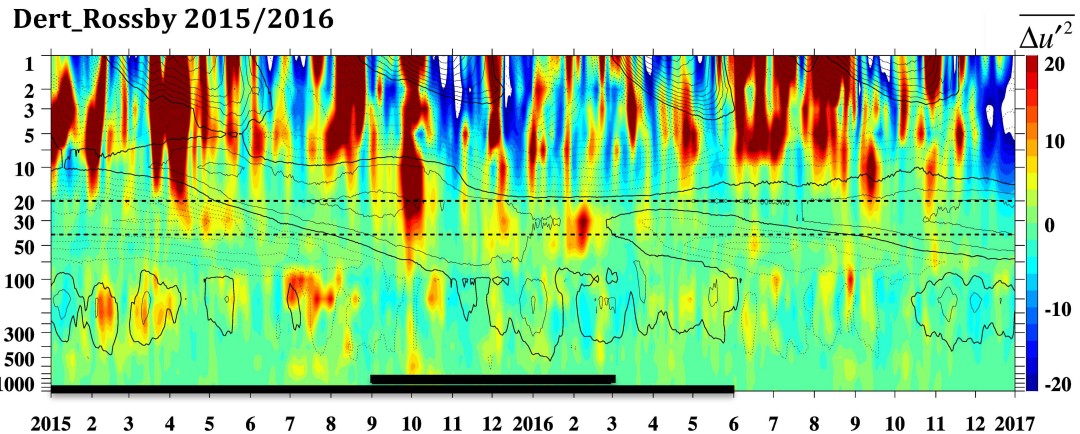

**Figure 2.** Temporal evolution of Rossby wave activity as squared zonal wind anomalies averaged over 10°S-10°N in the vertical direction. Color shading denotes the anomalies with respect to the monthly climatology (the unit is $m^2/s^2$). The zonal mean zonal wind is overlayed in black contours with contour interval of 5 $m/s$. The solid and dotted black contours denote westerly and easterly winds, respectively. The thick contours represent the zero wind line. The horizontal dashed lines denote the altitudes of 20 and 40 hPa. The horizontal solid lines denote the El Niño (single line) and strong El Niño periods (double line) in the bottom.

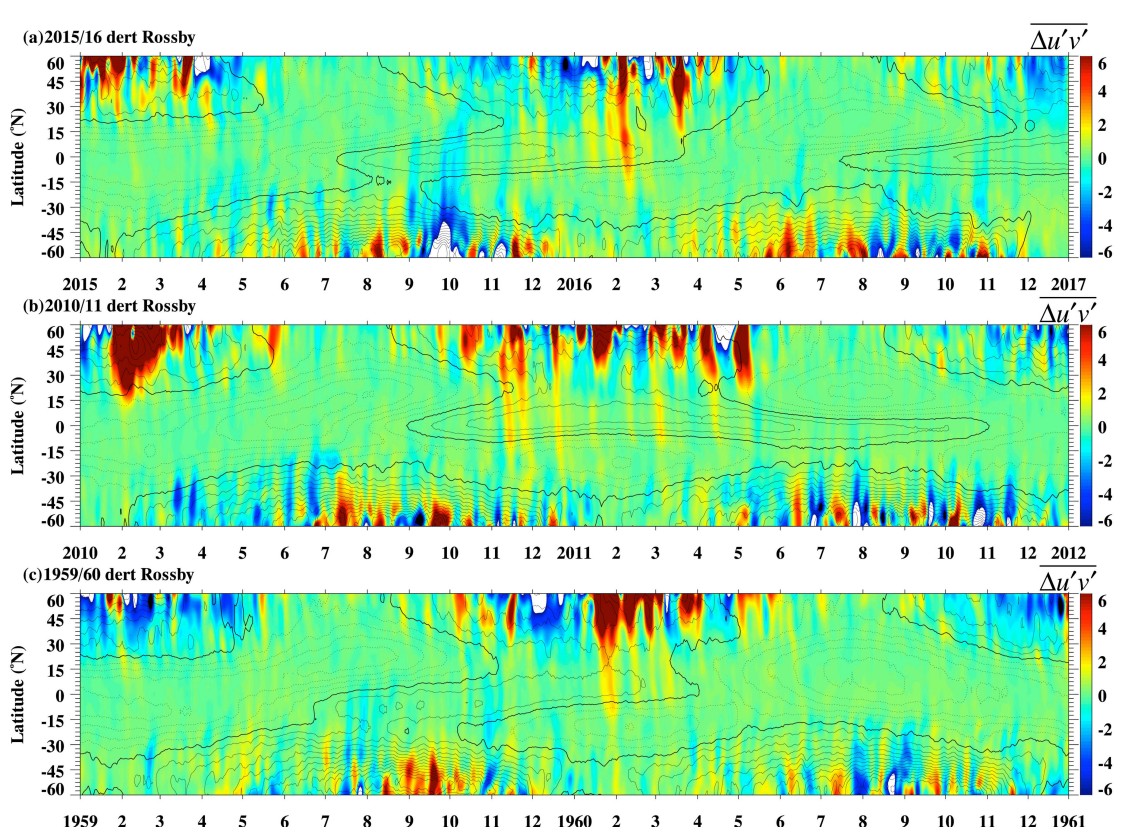

**Figure 3.** Temporal evolution of meridional propagation of westward propagating Rossby waves (with period of 5-70 days) horizontal momentum flux in (a) 2015/2016, (b) 2010/2011 and (c) 1959/1960 at 40 hPa. Color shadings denote the anomalies with respect to the monthly climatology (unit is $m^2/s^2$). The zonal mean zonal wind is overlayed in black contours with contour interval of 5 $m/s$. The solid and dotted black contours denote westerly and easterly winds, respectively. The thick contours represent the zero wind line.





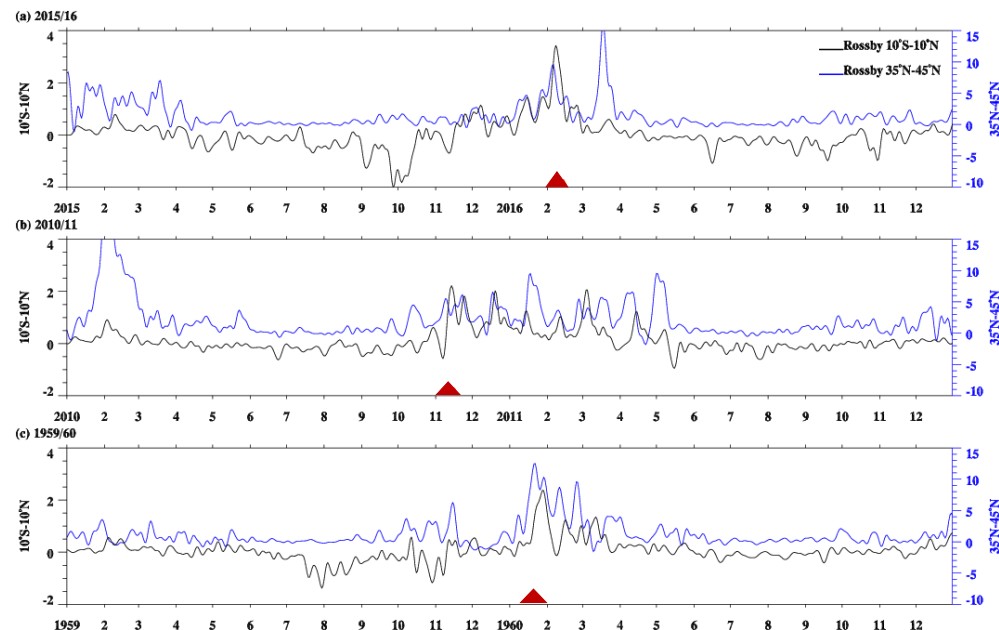

**Figure 4.** Temporal evolution of horizontal Rossby wave (with period of 5-70 days) momentum flux averaged over 10°S-10°N (black lines) and 35°N-45°N (blue lines) during (a) 2015/2016, (b) 2010/2011 and (c) 1959/1960 at 40 hPa. Red triangles in panels (a), (b) and (c) denote February 2016, November 2010 and January 1960, respectively.



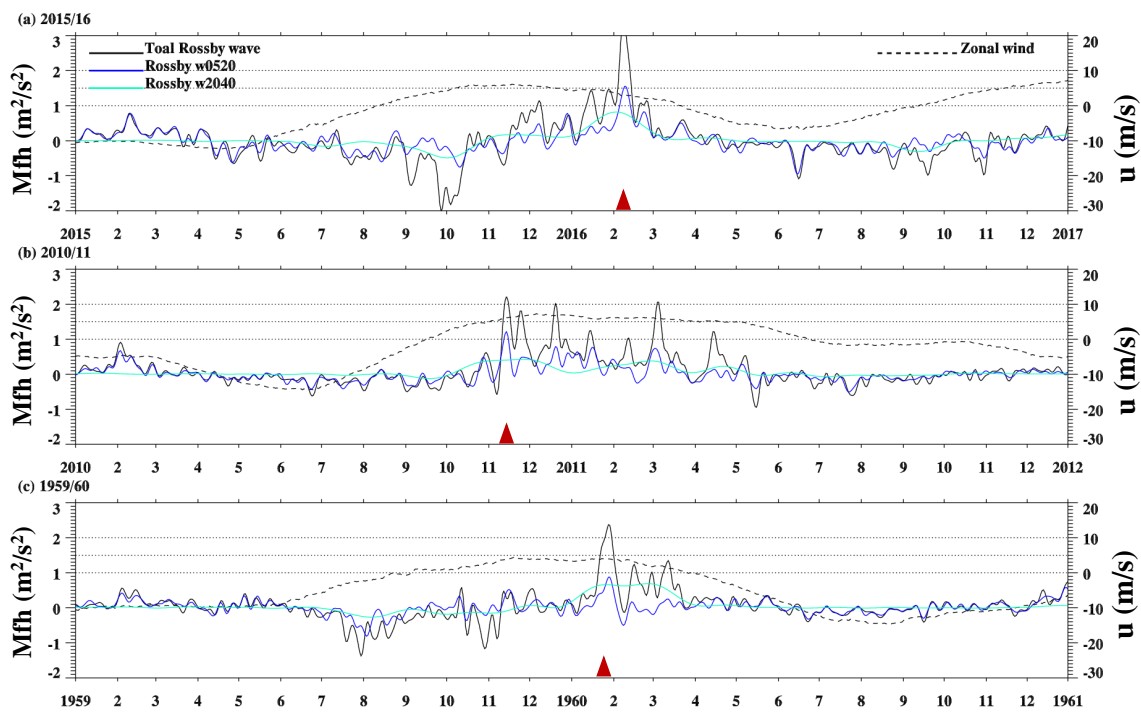

**Figure 5.** Temporal evolution of horizontal Rossby wave momentum fluxes over the tropics (10°S-10°N) during (a) 2015/2016, (b) 2010/2011 and (c) 1959/1960 at 40 hPa. The horizontal dotted lines denote the values of horizontal momentum flux of Rossby waves of 1 $m^2/s^2$, 1.5 $m^2/s^2$ and 2 $m^2/s^2$. The black, dark blue and light blue solid lines denote the horizontal momentum flux of total Rossby wave, Rossby w0520 and Rossby w2040, respectively. The dashed lines denote the background zonal wind during each case. Red triangles in panels (a), (b) and (c) denote February 2016, November 2010 and January 1960, respectively.

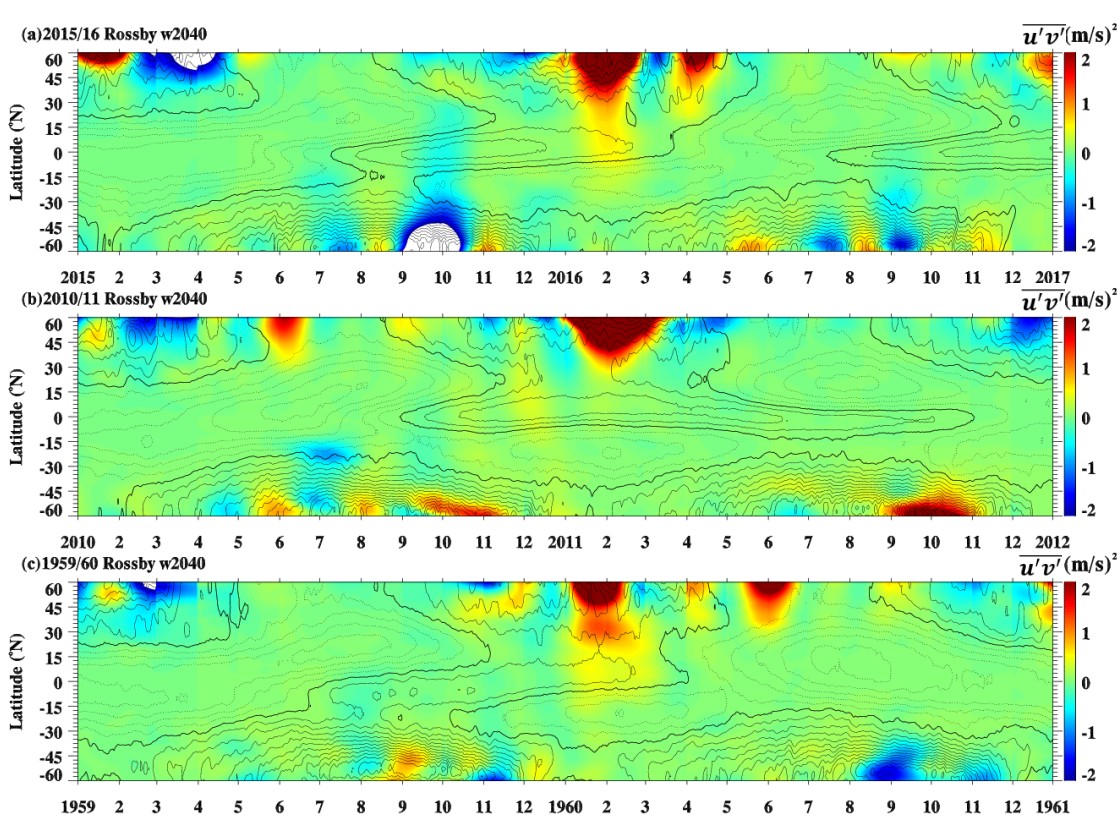

**Figure 6.** The time-latitude cross section of Rossby waves (with periods of 20-40 days and with wavenumber 1) horizontal momentum flux in (a) 2015/2016, (b) 2010/2011 and (c)1959/1960 at 40 hPa. Color shadings denote the anomalies with respect to the monthly climatology (0 $m^2 s^{-2}$). The zonal mean zonal wind is overlayed in black contours with contour interval of 5 $m/s$. The solid and dotted black contours denote westerly and easterly winds, respectively. The thick contours represent the zero wind line.



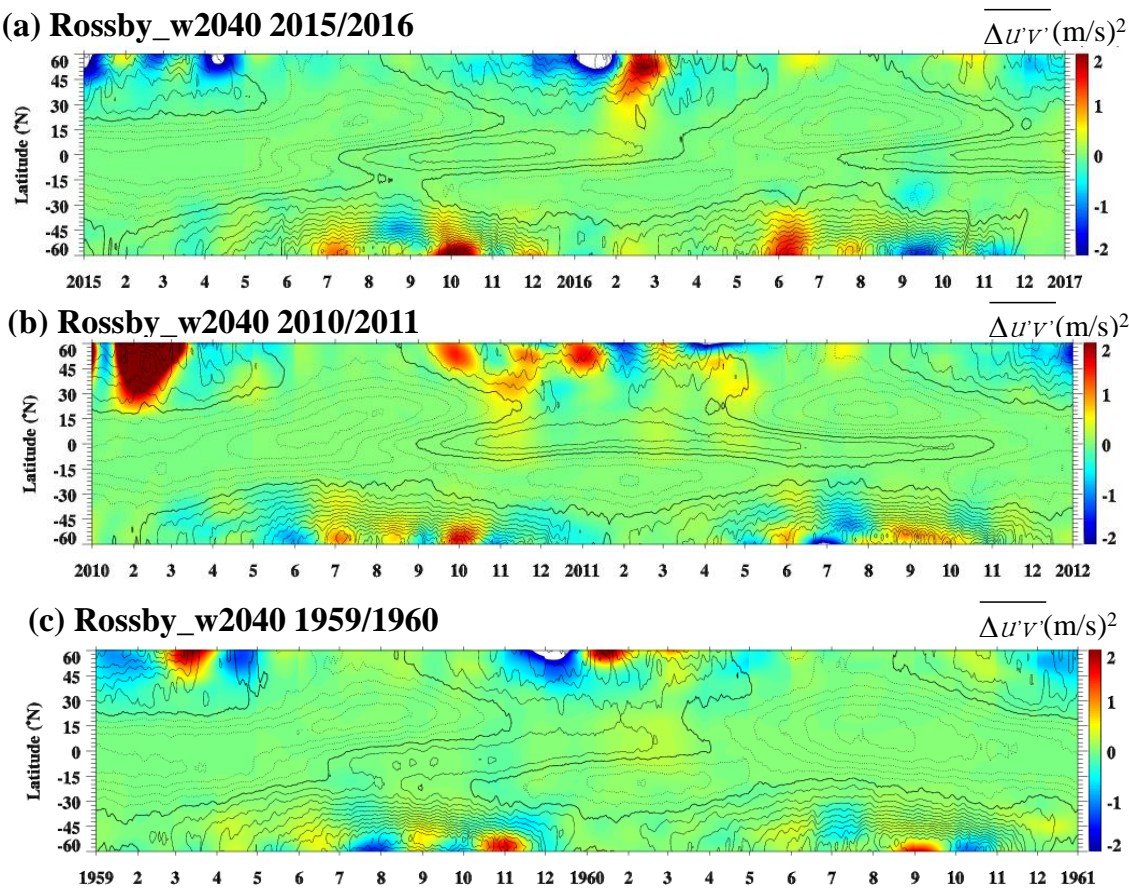

**Figure 7.** The latitude-time cross section of Rossby wave w2040 (with wavenumber 2) horizontal momentum fluxes in (a) 2015/2016, (b) 2010/2011 and (c) 1959/1960 at 40 hPa. Color shadings denote the anomalies with respect to the monthly climatology. The zonal mean zonal wind is overlayed in black contours with contour interval of 5 $m/s$. The solid and dotted black contours denote westerly and easterly winds, respectively. The thick contours represent the zero wind line.



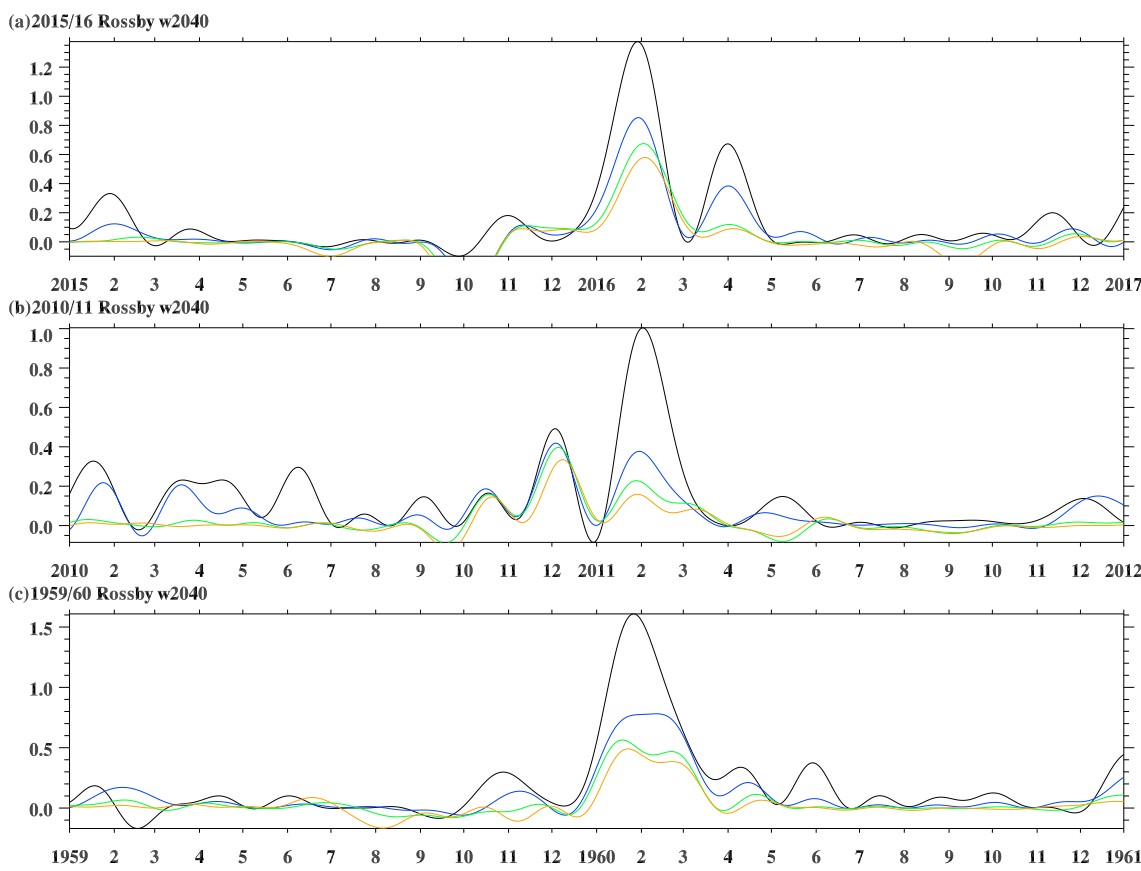

**Figure 8.** The temporal variation of Rossby w2040 (with wavenumber 1) horizontal momentum flux at 0°N (orange lines), 10°N (green lines), 20°N (blue lines) and 30°N (black lines) in 2015/2016, 2010/2011 and 1959/1960 at 40 hPa.



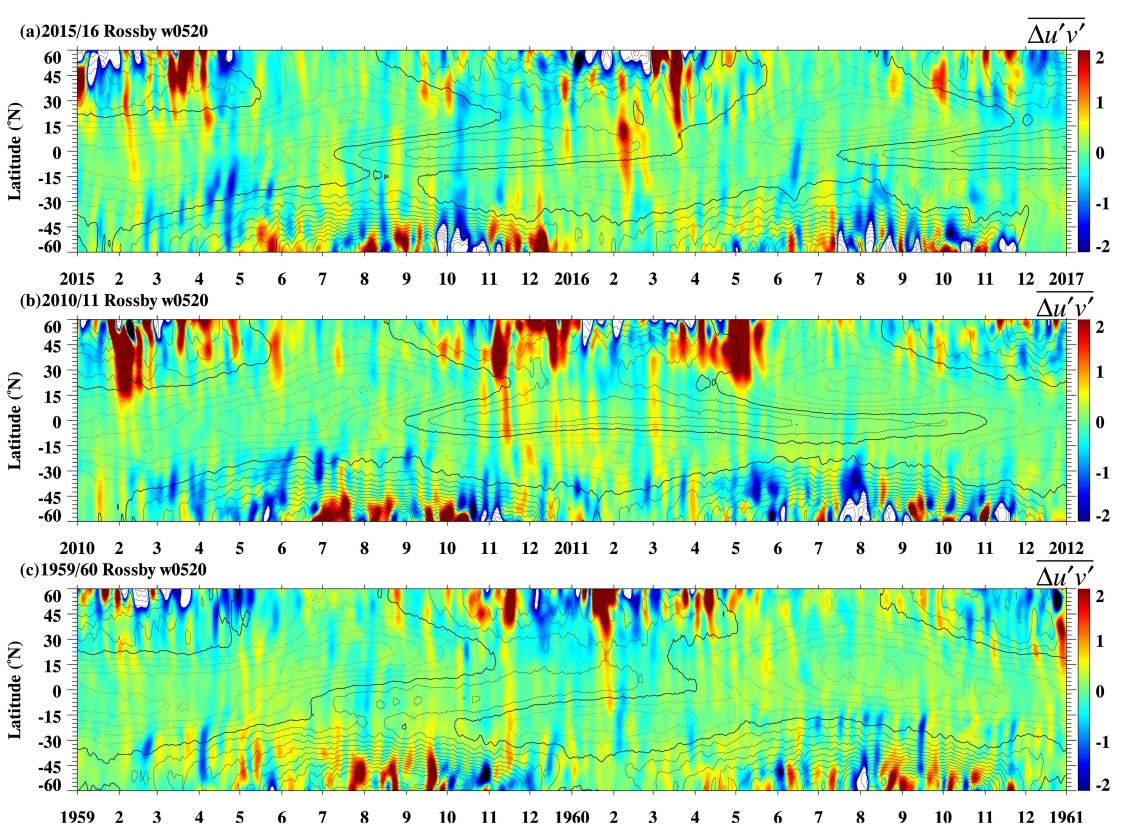

**Figure 9.** The time-latitude cross section of the Rossby waves (with periods of 5-20 days) in (a) 2015/2016, (b) 2010/2011 and (c) 1959/1960 at 40 hPa. Color shadings denote the anomalies with respect to the monthly climatology ($0\ m^2 s^{-2}$). The zonal mean zonal wind is overlayed in black contours with contour interval of $5\ m/s$. The solid and dotted black contours denote westerly and easterly winds, respectively. The thick contours represent the zero wind line.



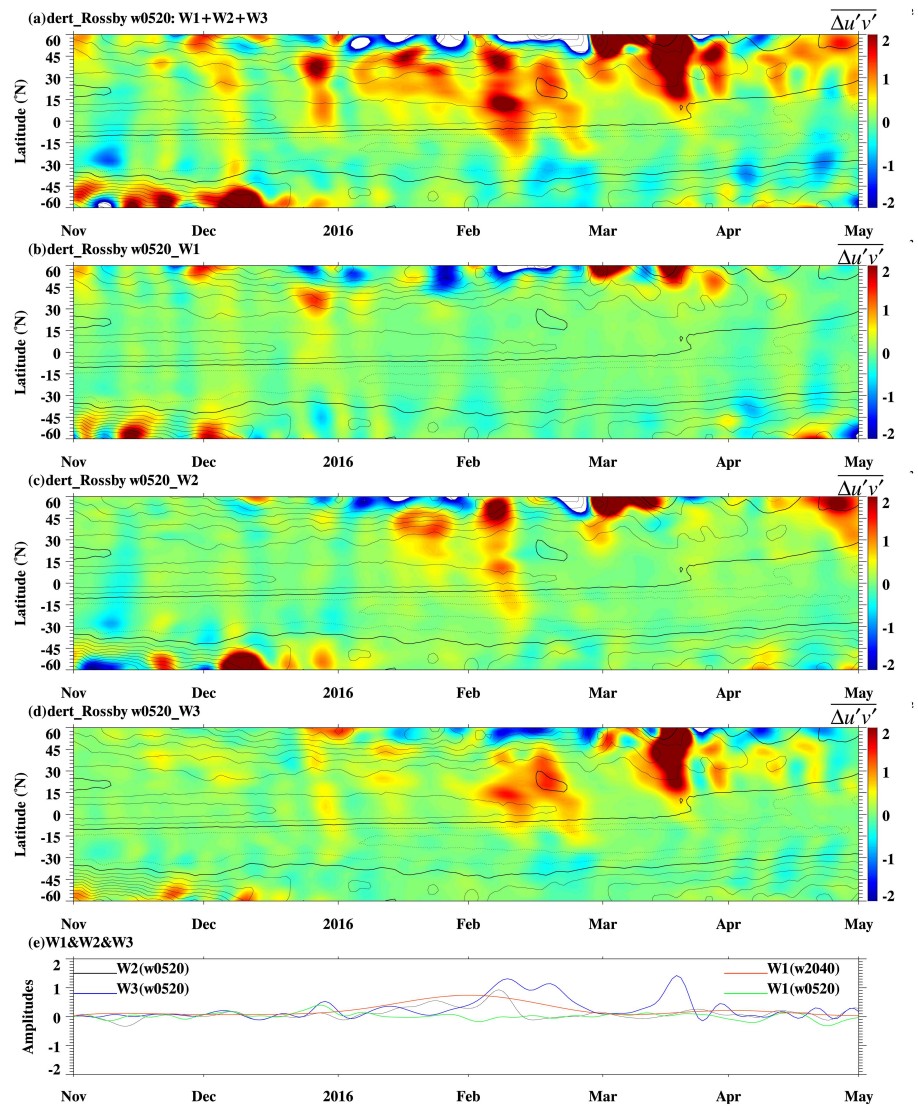

**Figure 10.** The time-latitude cross section of Rossby w0520 horizontal momentum fluxes (a) W1+W2+W3, (b) W1, (c) W2 and (d) W3 from November 2015 to April 2016. Panel (e) shows the horizontal momentum flux of W1, W2, W3 and Rossby w2040 (with wavenumber 1) at 40 hPa averaged from 10°N to 20°N. Color shadings denote the anomalies with respect to the monthly climatology (0 $m^2 s^{-2}$). The zonal mean zonal wind is overlayed in black contours with contour interval of 5 $m/s$. The solid and dotted black contours denote westerly and easterly winds, respectively. The thick contours represent the zero wind line. W1, W2 and W3 denote the Rossby w0520 with wavenumber 1, 2 and 3, respectively.

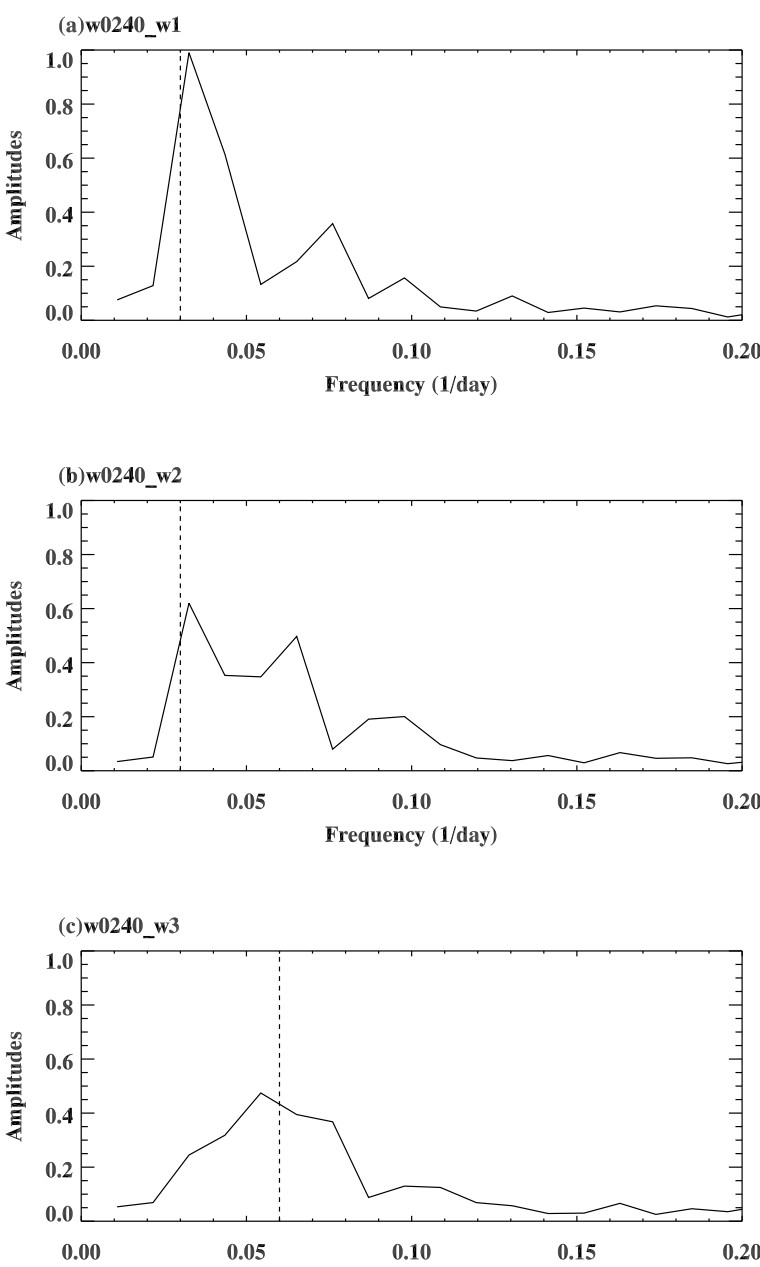

**Figure 11.** The spectrum of Rossby waves (with periods of 2-40 days) zonal wind perturbation averaged from 10°N to 20°N for (a) wavenumber 1, (b) wavenumber 2 and (c) wavenumber 3 at 40 hPa from January 2016 to Feburary 2016. The vertical dashed lines in panels (a) and (b) denote the frequency of 0.03 1/day. The vertical dashed lines in panel (c) denote the frequencies are 0.06 1/day.





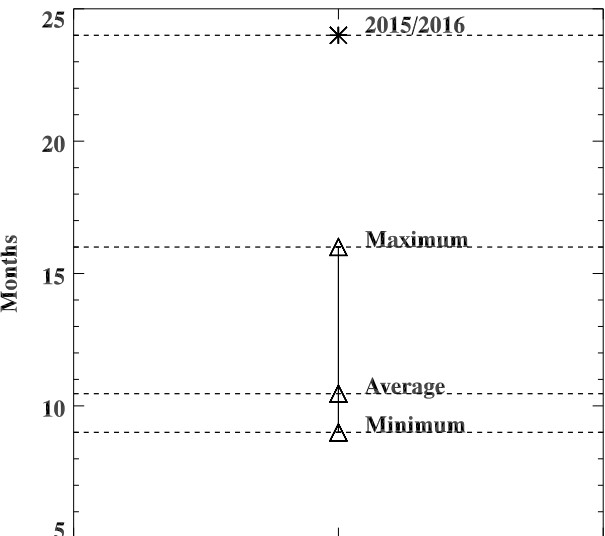

**Figure 12.** The minimum, average and maximum periods (triangles) of the westerly zonal wind at 20 hPa from 1958 to 2014. The star signal denotes the period of westerly zonal wind at 20 hPa in 2015/2016.

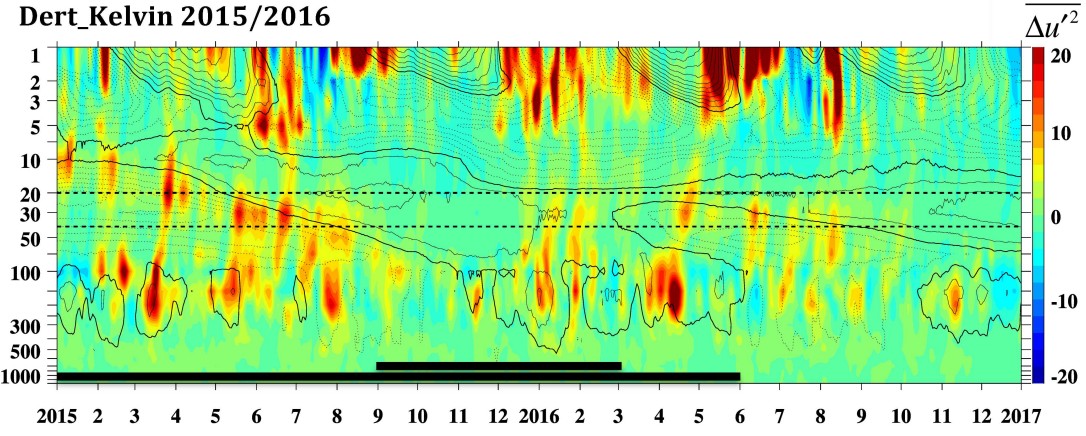

**Figure 13.** Temporal evolution of Kelvin wave activity as squared zonal wind anomalies averaged over 10°S-10°N in the vertical direction. Color shading denotes the anomalies with respect to the monthly climatology ($0\ m^2 s^{-2}$). The horizontal dashed lines denote the altitudes of 20 and 40 hPa. The zonal mean zonal wind is overlayed in black contours with contour interval of 5 $m/s$. The solid and dotted black contours denote westerly and easterly winds, respectively. The thick contours represent the zero wind line. The horizontal solid lines denote the El Niño (single line) and strong El Niño periods (double line) in the bottom.





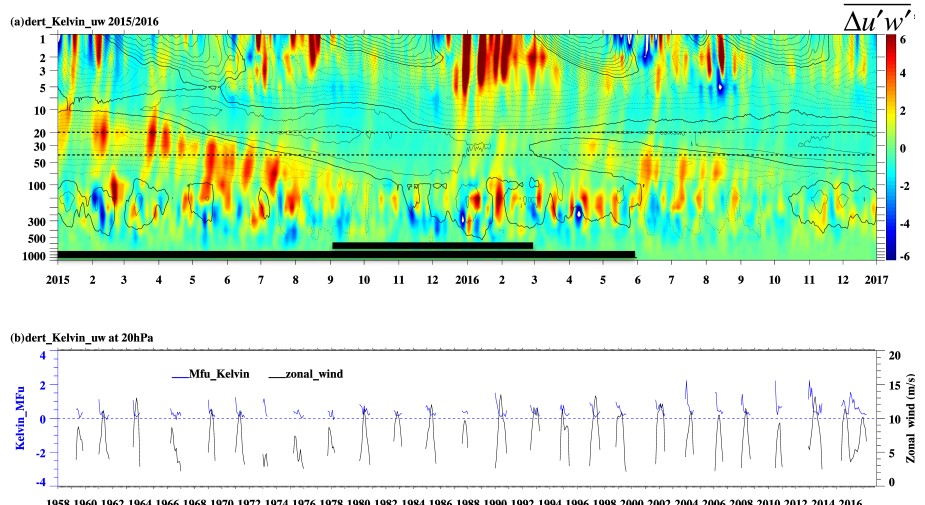

**Figure 14.** Temporal evolution of Kelvin wave vertical momentum fluxes averaged over 10°S-10°N (a) the daily result time-altitude cross section from 2015 to 2016 and (b) the monthly mean Kelvin wave vertical momentum fluxes (blue line) and the westerly background zonal wind (black line) at 20 hPa from 1958 to 2017. Color shading denotes the anomalies with respect to the monthly climatology (unit is $10^{-3}$ $m^2 s^{-2}$). The zonal mean zonal wind is overlayed in black contours with contour interval of 5 $m/s$. The solid and dotted black contours denote westerly and easterly winds, respectively. The thick contours represent the zero wind line. In panel (a), the horizontal dashed lines denote the altitudes of 20 hPa and 40 hPa, respectively. The horizontal solid lines denote the El Niño period (single line) and strong El Niño period (double lines) in the bottom. In panel (b), the dashed line denotes the vertical Kelvin wave momentum flux value of 0 $m^2 s^{-2}$.

## Kelvin u² anomalies at 200 hPa

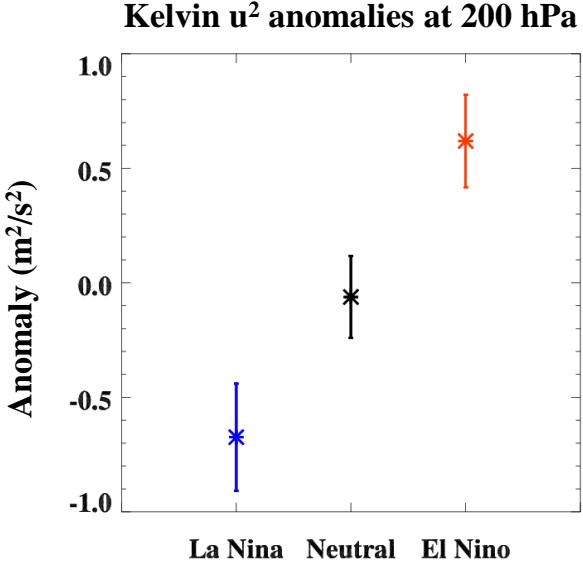

**Figure 15.** The squared perturbations in zonal mean zonal wind for Kelvin waves averaged over 10°S-10°N at 200 hPa for La Niña (blue star), Neutral (black star) and El Niño (red star) conditions. The error bars denote +- 1 standard deviation of the mean values.