# Peer review of "On the forcings of the unusual QBO structure in February 2016"

_Atmospheric Chemistry and Physics, 2019_

## Referee Comment (RC1) · Anonymous Referee #1 · 27 Sep 2019

General Comments: The authors present a study of Rossby and Kelvin wave activity as it relates to the 2015/16 QBO disruption. The analysis is detailed, and describes the evolution and impact of various Rossby wave modes organized by wavenumber and/or frequency. Although the wind structure was conducive to the disruption in other select cases, one particular feature marking the disruption as anomalous is reported to be a resonance between wavenumber 1 and 2 Rossby modes originating in the extratropics, which combine to generate a wave 3 mode in the subtropics that provides significant easterly acceleration near 40 hPa. The authors also studied the prolonged westerly phase near 20 hPa. They provide evidence that, in the case of the disruption, a record-strength El Nino directly increased Kelvin wave activity, which caused the prolonging.

Overall, the paper is well-written with strong scientific justifications. It provides a

uniquely detailed look at the wave activity surrounding the QBO disruption. I have just some minor concerns with specific results, which I have written below.

Specific Comments: Line 268 - All components of a Rossby wave packet do not necessarily travel together since the Rossby wave is dispersive.

Line 300 - It seems somewhat contrived to choose the 0.066/day peak to discuss the W3 being locally generated when that is not the frequency with the strongest power. Where does the power at 0.055/day come from?

Line 303 - I am also somewhat unclear on the following: you claim that a combination of the fast W2 (momentum flux Fig. 10c) and slow W1 (amplitude Fig. 10e) create the slow W3. When you say fast W2 are you talking about the w0520 Rossby wave? The 0.033/day frequency implies a period of 30 days, which is inconsistent with your "fast W2" resonance theory. Can you elaborate?

Line 350- Enhanced with respect to what? The previous winter? Of course the Kelvin wave activity depends on the wind structure. There are no Kelvin waves here until February because the winds are too strong westerly. The general tropospheric Kelvin wave activity looks to be lower than the previous winter too.

Line 356 - Again, is the Kelvin wave momentum flux "enhanced" just because Kelvin waves can propagate into this region of weak westerlies? If yes, this is not surprising.

Line 360 - This is not necessarily true. Kelvin waves can propagate through westerly flow as long as their phase speed is greater than the flow speed. So the fact that the westerlies are weak in this case allows them to do so.

Line 374 - In the standard QBO paradigm, increased Kelvin wave activity causes descent of the westerlies, because the waves accelerate the flow below their critical level, which in turn lowers the critical level, and so on. Why in this case does the increased Kelvin wave activity not promote descent?

Line 394 - There are a few studies to have linked Kelvin wave activity to El

Nino. cf. Yang and Hoskins (https://doi.org/10.1175/JAS-D-13-081.1), Das and Pan (https://doi.org/10.1016/j.scitotenv.2015.12.009), Rakhman et al (doi:10.1088/1755-1315/54/1/012035)

Technical Corrections: Figure 1 caption - Please delete the first instance of "The vertical blue lines denote the time periods..." in the first sentence (it is repeated later). In panel (a) there are no red triangles, so either add them or do not mention them in the penultimate sentence.

Figure 5 - The label should read "Total Rossby wave" (typo)

Figure 8 - Can you add a legend to this figure? Also, are the time axes correct? You mention on Line 241 different months for the peaks in each year, but these all peak in February.

Figure 10 caption - Please move the sentence about Panel (e) to the end, the descriptions that follow it all refer to the other panels so it is a little confusing.

Figure 11 caption - "February" (typo)

---

## Referee Comment (RC2) · Anonymous Referee #2 · 28 Sep 2019

"On the forcings of the unusual QBO structure in February 2016", written by Li et al.

This is very interesting manuscript, detailing the wave forcing of the unusual QBO event in 2015/2016 boreal winter. This paper attempts to investigate the responsible waves and mechanisms for 1) the reversal event at 40 hPa and 2) unusually long westerly zonal mean zonal wind at 20 hPa by separating the contributions of individual wavenumbers and different timescales, and using fields from the combined ERA-40 and ERA-Interim reanalyses.

I find a performed analysis clear and convincing, supporting main augments and findings of this manuscript. Although I am recommending major revisions, most of what I am requesting will not involve much if any in the way of new calculations, but rather requires some editing for clarity and conciseness as well as more interpretation which

would require changes in the text and some addition of background and comparison with other studies. The results are potentially useful to the community and I think it will be a very valuable contribution to the literature once some clarifications and edits are made to both the arguments and the methodology.

General comments: First of all, this manuscript can benefit from some revisions and editing to improve for clarity and conciseness. Some recommendations are provided in the comments below. As everyone's writing style is different, those comments are by no means requirement but rather indications where writing could benefit from some editing to make your massage clearer. Second, some clarifications should be done in Data and Methods section. Line by line comments are provided below. For example, I strongly recommend to rewrite "Wave filtering" subsection in a more clear way as it can be very confusing to some. In several places, your statements like "It is difficult to (do something) " can be confusing as it is not very clear if you've applied your technique to the dataset. How did you do your filtering or any other analysis step by step? State these steps first without any unnecessary information and then provide any relevant comments regarding these techniques at the end. See more detailed comments below. Furthermore, Introduction or/and Discussions are lacking a discussion of literature on relationship between ENSO and Kelvin waves. While a role of enhanced Kelvin Wave activity due to strong 2015/2016 El Nino event in producing the extended westerly zonal wind near 20 hPa is definitely new (and very interesting) result, the enhanced Kelvin wave activity and its relation to ENSO definitely was previously looked at and documented. For instance paper by Yang and Hoskins (2013) along with others should be mentioned and cited. [Yang, G. and B. Hoskins, 2013: ENSO Impact on Kelvin Waves and Associated Tropical Convection. J. Atmos. Sci., 70, 3513–3532, https://doi.org/10.1175/JAS-D-13-081.1]

Specific comments:

Line 95: Why is combined ERA-40 and ERA-Interim reanalyses are used instead of just using ERA-40 or ERA-Interim reanalyses by itself. It would be useful for readers

to see a short statement explaining benefits and limitations of individual datasets vs combined one.

Line 96: Using layer averaging (30-50 hPa) to represent 40hPa level is fine if 40 hPa level is not available as an output. However, this (or other reasons for layer averaging) should be mentioned.

Line 106: "similar to U40". Is U20 also defined as a layer averaged U between two pressure levels? Please clarify.

Line 115-117: "Previous studies . . ." How this sentence connects to the filtering techniques you are discussing? This seems an odd place for comparison with previous studies unless you are comparing techniques that you are using in this study with one's in previous studies.

Line 118: "following dispersion curve" Is it a part of the "kf-filter" function technique or something you performed after applying this filter. If it is unrelated to the "kf-filter" than Line 112 shouldn't be a fist sentence of this paragraph.

Line 118, also line 124 A use of statements like "it is difficult . . ." or "we can .." are very confusing and distracting since I am not sure if you applied the technique to your data or just speculating about it. Make sure you are clearly explaining your methods.

Line 173: "was weaker". What do you mean by this (its amplitude or persistence in time)? To my eye these three events don't stand out very well when compared to other years in this record but I could be missing something. Being specific may help.

Line 184: "However, we cannot . . . " very wordy sentence. How about something like this : "Since enhanced Rossby wave behavior doesn't extend to the lower and higher levels, the observed signal less likely originated from vertical propagation." Side note: Please comment a strong signal in enhanced Rossby wave activity during October 2015.

Line 224: What do you mean under "it's different components". What components

exactly do you mean? Be more specific.

Line 236: What do you mean by "weaker but similar"? These are two contradictive words. Weaker in strength (or amplitude) but similar in shape?

Line 245: "slower than 0.15 cpd, or 6-7 day frequency". Should it be day-1 ? Also, can you change these to a more familiar timescales which you are using (in days)? This will make comparison with your study easier and you'll support your augment for their study to include both, the faster and quasi-stationary Rossby waves". Otherwise it is not very obvious.

Line 325: Very confusing. Please rewrite. First sentence of the paragraph should reflect its main idea or statement.

Line 351: "50 to 20 hPa" Do you mean "50 -200 hPa" because I see a strong signal in the upper troposphere as well?

Line 384 and also 393-394: As mentioned in General Comments, this is definitely not a new result. Previous studies, discussing enhanced KW activity during El Niño events, have to be cited and also discussed. One of these studies is by Yang and Hoskins (2013). Yang, G. and B. Hoskins, 2013: ENSO Impact on Kelvin Waves and Associated Tropical Convection. J. Atmos. Sci., 70, 3513–3532, https://doi.org/10.1175/JAS-D-13-081.1

Technical comments:

Line 136: "In order to explore" -> "To explore" (remove "In order")

Line 152: "We explore a possible . . .. . . in section 3.1.3". -> remove this sentence

Line 130: "Figure 2 illustrates . . .. ." Remove entire sentence since the same information is included in Figure 2 caption.

Line 183: "Enhanced Rossby wave . . . in the NH" -> remove the entire sentence (it doesn't have a logical connection with sentences before and after)

[Figure]

Line 202: "Figure 4 shows that the horizontal .." -> "As expected, the horizontal Rossby wave momentum fluxes peak in the . . ."

Line 211: "We calculated the mean . . .". -> "The mean horizontal . . . in the tropics are shown in Figure 5. The maximum . . ."

Line 223-224 "Their study was . . ." -> "Although their study . . . (35.8 hPa and 40 hPa respectively), we describe the evolution . . ."

Line 229-230: Remove "As shown in Figure 5" and connect it with next sentence -> "The Rossby waves . . . (red triangles in Figure 5) with horizontal momentum flux having maximum values of about . . . ."

Line 233: "We calculated . . . " -> "To explore the sources of quasi-stationary Rossby waves, Figures 6 and 7 show the horizontal . . . ."

Line 234: remove entire sentence "Then we analyzed . . ."

Line 235-237: Remove "Figures 6 and 7 show" and combine this sentence with a next one -> "The contribution of Rossby waves . . .. in November 2010, while . . . "

Line 239: Put a dot after "in Figure 8", remove "which reveals that"

Line 240-242: change to : "The peaks of Rossby waves w2040 in Jan 1960, Nov 2010 and Feb 2016 at 40 hPa occurred earlier at higher latitudes, which is indicative of their extratropical origins. This is in agreement with Figures 5, 6, and 7 revealing an important contribution of quasi-stationary Rossby waves to the enhanced Rossby activity in the tropics mainly by wavenumber 1 and to a lesser degree by wavenumber 2 of extratropical origin."

Line 252-260: Need some revisions for conciseness. For example, there are few sentences that can be combined together into one and extra words have to be removed (e.g., "as shown in" (line 256), "the results reveal that in particular" and so on).

Line 284: How about : "The meridional gradients are negative in late March 2016,

which indicates barotropic – baroclinic instability (see section 2.4). In this study, W3 maximizes one month prior to March 2016. This is consistent with Coy et al. (2017) . . .

Line 341: Start this sentence with "The average period . . ." and remove all words before it ("By performing . . . we found that").

Line 342: Add -> "(Figure 12). However, in 2015/2016 the . . ."

Line 351: Remove "Figure 13 shows that . . .pertubations had" -> "from 50 to 200 hPa with maximum amplitudes . . ."

Line 380: remove parenthesis around 20 hPa

Figures: It would be great if you can increase thickness of the zonal mean zonal wind contours.

---

## Referee Comment (RC3) · Anonymous Referee #3 · 8 Oct 2019

This is an interesting study that sheds more light on the dynamics that led to the anomalous QBO structure in early 2016. By spectral filtering horizontal momentum flux of Rossby waves and vertical momentum flux of Kelvin waves is deduced for the years 1958-2017 using the ERA-40 and ERA-Interim reanalyses. In agreement with previous work, it is found that the anomalous QBO westerlies are forced by Rossby waves from the extratropics. The three known cases of strong extratropical Rossby wave forcing during QBO easterly phases (1959/1960, 2010/2011, and 2015/2016) are compared. What is new and exciting in Lin et al. is the attribution of the anomalous QBO forcing to different regimes of Rossby waves. It is shown that Rossby waves of 5-20days period play an important role, and different from the 1959/1960 and 2010/2011 events, a wavenumber 3 Rossby wave is generated at 15N by wave wave interaction of

quasi-stationary waves 1 and 2. This wave also seems to play an important role in the anomalous forcing. The extended period of QBO westerlies is attributed to enhanced Kelvin wave activity caused by El Nino.

Overall, the paper is well written and of relevance for the readership of ACP. Publication is therefore recommended after addressing my major comments. Addressing these comments should however be straightforward.

————————————————————————————————

Major comments:

(1) Usually, to capture the propagation and interaction of global scale waves Eliassen Palm (EP) fluxes are calculated. You are calculating only  (vertical momentum flux) and  (horizontal momentum flux) which are only part of the EP flux and do not mention the limitations of this approach.

(2) The whole study is based on the kf-filter method. However, almost no information is given how this method was applied.

————————————————————————————————

Specific comments:

(1) General comment: When discussing momentum fluxes, you often do not state clearly what you are discussing - the momentum fluxes, or their anomaly. This happens in the text, as well as in the figure captions and in the figure legends! In my comments, I mentioned only a few occurrences because they are too many. Please revise carefully throughout!

(2) General comment: 20hPa (about 27.5 km altitude) is in the middle stratosphere, not the upper stratosphere. The stratosphere extends from about 10-20km to about 50km. Please revise carefully throughout the paper!

(3) p.2, l.44-46 here you write: The eastward propagating Kelvin waves provide the

main eastward acceleration for the initiation of the QBO westerly phase. In contrast, the westward propagating Rossby waves provide the main westward acceleration for the initiation of the QBO easterly phase.

This is not entirely correct because also small scale gravity waves contribute. Mainly both Kelvin waves and small scale gravity waves contribute to the forcing of the QBO westerly phase, while global scale westward traveling tropical waves and small scale gravity waves contribute to the forcing of the QBO easterly phase (see for example Ern and Preusse, GRL, 2009 and Ern et al., JGR, 2014, Garcia and Richter, JAS, 2019).

References:

Ern, M., and P. Preusse, Quantification of the contribution of equatorial Kelvin waves to the QBO wind reversal in the stratosphere, Geophys. Res. Lett., 36, L21801, doi:10.1029/2009GL040493, 2009.

Ern, M., F. Ploeger, P. Preusse, J. C. Gille, L. J. Gray, S. Kalisch, M. G. Mlynczak, J. M. Russell III, and M. Riese, Interaction of gravity waves with the QBO: A satellite perspective, J. Geophys. Res. Atmos., 119, 2329-2355, doi:10.1002/2013JD020731, 2014.

Garcia, R. R., and J. H. Richter, On the Momentum Budget of the Quasi-Biennial Oscillation in the Whole Atmosphere Community Climate Model, J. Atmos. Sci., 76, 69-87, 2019.

(4) p.3, l.68: These references address only global scale waves. A reference for small-scale gravity waves should be added. Using model simulations, differences in the QBO forcing by gravity waves for different ENSO conditions were investigated, for example, by Kang et al., JAS, 2018.

Reference:

Kang, M.-J., H.-Y. Chun, Y.-H. Kim, P. Preusse, and M. Ern, Momentum Flux of Convective Gravity Waves Derived from an Offline Gravity Wave Parameterization. Part II:

Impacts on the Quasi-Biennial Oscillation, J. Atmos. Sci., 75, 3753-3775, 2018.

(5) p.3, l.91: Please state more clearly that a merged ERA-40/ERA-Interim dataset is used to have a longer time series because ERA-Interim starts with the year 1979 and does not cover the earlier period.

(6) p.4, l.114/115: You should elaborate much more on the kf-filter! As far as I understand, for a fixed latitude and altitude, you enter the whole time series of 60 years with a longitude resolution of 2.5 deg into the kf-filter. The time series is tapered to zero at both ends. Which function is used? Split-cosine-bell as stated in https://www.ncl.ucar.edu/Document/Functions/Built-in/taper.shtml ? Did you use the standard settings of the taper of p=0.1? In this case, you would have to discard 3 years at each end of the time series and valid data are obtained only for 1961-2014. This would contradict your statement on p.5, l.139 that only the years 1958 and 2017 would be affected. Please explain!

(7) p.5, about section 2.3: Please explain why you calculate momentum fluxes in this way! Possibly, you are missing something or biasing your analysis by not calculating the full EP flux vector! Therefore at least the limitations of your approach should be clearly stated.

Usually, for global scale waves EP fluxes are calculated to capture their propagation and effect on the background flow, however you are neglecting heat fluxes.

By calculating  you are using the quasi-geostrophic approximation of the EP flux in meridional direction (see for example Matthias and Ern, ACP, 2018). This approximation can be used for extratropical Rossby waves and should be sufficient for diagnosing the meridional propagation direction of Rossby waves from the extratropics. In the tropics, however, wave motions can become ageostrophic, and this approximation may no longer hold. This is the case for tropical waves like Kelvin waves or tropical Rossby waves.

The term  is part of the vertical component of the EP flux and should be a a good approximation for Kelvin waves because v' is zero for these waves.

Reference: Matthias, V., and M. Ern, On the origin of the mesospheric quasi-stationary planetary waves in the unusual Arctic winter 2015/2016, Atmos. Chem. Phys., 18, 4803-4815, 2018.

(8) p.5, l.149/150: here you state that Rossby waves would not contribute much to  as could be seen from Figs. S1 and S2 in the supplement. In the supplement, however, only "anomalies" of  and  from their monthly means are shown.

(9) captions of Figs. 3, S1 and S2: First you write that horizontal momentum fluxes would be shown, and later in the caption you write that color shadings would be anomalies with respect to the monthly climatology. This is very confusing, please write more clearly!

(10) Follow-up question: Were the monthly climatologies temporally interpolated to single days to avoid jumps in the anomalies from one month to the other? Please clarify!

(11) p.5, l.152/153 and l.155: This is not entirely correct: Shuckburgh et al. (2001) investigated only barotropic instabilities, not baroclinic instabilities! Only Coy et al. (2017) included also baroclinic instabilities.

(12) p.7, l.210: Why did you select w2040 for the quasi-stationary Rossby waves, and not w2070? Choosing w2070 would be much more intuitive!

(13) p.7, l.214-219: Would also the contributions of w2070 be similar in all three cases? Please note that w0520 and w2040 do NOT sum up to the "total Rossby" fluxes! Possibly w4070 might show also considerable case to case variability.

(14) p.10, l.305: here you state: "the quasi-stationary W1 and the faster W2 which came from the extratropics, generated W3 locally..."

[Figure]

This contradicts your statements from above that W3 was generated by W1 and W2 of 0.033 dayˆ-1 which are both in your quasi-stationary range of frequencies. Please clarify!

(15) p.10, l.313/314: here you state that: "W3 cannot be an equatorial wave mode in principle, otherwise its amplitude would be by definition maximizing at the equator."

I think that this statement does not generally hold. Equatorial Rossby modes can be symmetric or anti-symmetric with respect to the equator. Therefore either u' or v' could be zero at the equator, leading to zero . For a survey of equatorial modes see, for example, Yang et al., JAS, 2003, their Fig.3.

Reference: Yang, G.-Y., B. Hoskins, and J. Slingo, Convectively Coupled Equatorial Waves: A New Methodology for Identifying Wave Structures in Observational Data, J. Atmos. Sci., 60, 1637-1654, 2003.

(16) p.10, l.325: Enhanced momentum flux does not necessarily mean that the zonal wind is accelerated or decelerated. There is only an effect on the background flow when this momentum is deposited (if there is a non-zero divergence of the EP-flux). Of course, enhanced momentum fluxes can lead to stronger EP-flux divergences.

(17) About Figs.13 and 14: As Kelvin waves in the stratosphere are modulated rather by the QBO than by a seasonal cycle, does it really make sense to show deviations of Kelvin wave amplitudes or momentum flux from a monthly mean climatology?

(18) p.12, l.360/361: Another possibility could be that these Kelvin waves have phase speeds that exceed the westerly wind. As can be seen, for example, in Ern et al., ACP, 2008, Kelvin waves with high phase speeds are not much modulated by the QBO, while slower phase speed Kelvin waves are strongly modulated by the QBO with minimum amplitudes during westerly winds.

Reference: Ern, M., P. Preusse, M. Krebsbach, M. G. Mlynczak, and J. M. Russell III, Equatorial wave analysis from SABER and ECMWF temperatures, Atmos. Chem.

[Figure]

Phys., 8, 845-869, 2008.

(19) p.13, l.417/418: This is not correct: Both W1 and W2 have the same frequency and fall in the frequency range that you call quasi-stationary.
* * *
Other comments:

(1) p.2, l.35: The westerly mean flow in the tropical stratosphere generally favors -> If the mean flow in the tropical stratosphere is westerly, it generally favors

(2) p.5, l.150: Fig.12a does not exist! Should this read Fig. S1? Please check!

(3) Caption of Fig.1, l.2: ??? horizontal Rossby wave momentum flux -> Rossby wave horizontal momentum flux anomaly

(4) Caption of Fig.1: Different from what is stated in the caption, there are no red triangles in Fig.1a

(5) p.6, l.180 / Fig.2: It is unclear what is shown! Probably your notation is misleading! In the figure legend of Fig.2 it reads: $<du'^2>$, suggesting that you calculate a climatological u' distribution, and you are showing deviations from that average u' for a particular period. In this case, however, there could be no negative values because values are squared for display! So my guess is that it should read $d$ in the figure legend. Similar, in Fig.3 it should probably read $d$ instead of $<du'v'>$.

(6) p.6, l.190: horizontal Rossby waves momentum flux -> Rossby wave horizontal momentum flux anomalies

(7) Fig.5: Please mention in the figure caption that "total Rossby wave" corresponds to w0570.

(8) p.8, l.251: momentum flux -> momentum flux anomaly

(9) p.8, l.251, suggested rewording, as this cannot be seen from Fig.9: The contribution
* * *
Interactive
comment

of Rossby waves w0520 is largest in February -> In the tropics, the largest anomaly of Rossby waves w0520 is found in February

(10) p.8, l.252: momentum flux was stronger -> momentum flux anomaly was stronger

(11) caption of Fig.10, l.1: horizontal momentum fluxes -> horizontal momentum flux anomalies

(12) Caption of Figs. S5 and S6: Please mention the altitude/pressure of these sections.

(13) p.9, l.283: and below -> and at higher altitudes

(14) p.9, l.284: period below 40 hPa. -> period at pressures below 40 hPa.

(15) p.9, l.284: The meridional gradient starts -> At 40 hPa the meridional gradient starts

(16) p.9, l.285: barotropic and baroclinic -> barotropic and/or baroclinic

(17) p.10, l.298: W3 has the most complex peaks, the stronger peaks are corresponding to the frequencies are at -> W3 has a broad peak with the strongest contributions at

(18) In Fig.14b: why are the lines interrupted? Are no-shows insignificant values? If yes, please state in the figure caption!

---

## Author Comment (AC1) · 31 Jan 2020

**Response to reviews of "On the forcings of the unusual QBO structure in February 2016" by Haiyan Li et al.**

Dear Editor,

We would like to thank the three anonymous reviewers for their comments and suggestions, which led to a substantial improvement of the manuscript's quality. In our 'General comments from the authors' (below), the most significant changes made in the manuscript are grouped and explained in more detail.

In the following paragraphs, we include our *point-by-point responses (in dark gray)* to each **reviewer comment (bold black)**.

In the revised manuscript with tracked changes, minor and technical revisions are marked in blue, whereas more substantial changes are highlighted in yellow.

Yours sincerely,
Haiyan Li
Robin Pilch Kedzierski
Katja Matthes

**General comments from the authors:**

In going through all reviewer's comments, we identified three main issues in our manuscript:

**1)** Clarity of 'Wave filtering' section (2.2), justification and limitations of the 'Momentum flux calculation' section (2.3):

Significant parts of sections 2.2 and 2.3 have been rewritten. In section 2.2 we prioritized clarity about the options the kf-filter has, as well as the use the use given to them. In section 2.3 we specify that ours is a simplified approach, stating its limitations while reassuring the reader that the method is sufficient for the purposes of the study. Here we particularly appreciate the reasoning and reference suggestion by reviewer #3.

**2)** The wave resonance mechanism: reviewers #1 and #3 pointed out inconsistencies in our interpretation of Fig. 11 and attributing specific peaks in the frequency spectra to the resonating W1 and W2 of extratropical origin.

We didn't realize these inconsistencies before, and as the reviewers pointed out the interpretation of the resonance mechanism wasn't in line with the results from Figs. 10 and 11.

We've performed a more detailed analysis of the frequencies of W1-3, and it turned out that all had varying speeds/frequencies during the Feb. 2016 event (see the updated Fig.11 and related discussion in section 3.1.3 in the new manuscript version). Therefore individual peaks in the frequency spectra of the previous manuscript version didn't really correspond to the W1-3 properties during the resonance event. The more detailed results presented now support the main conclusions in a more robust way.

**3)** Lack of literature and discussion about ENSO and Kelvin waves:

      We apologize for this, we should have looked more into this topic before completing the submitted manuscript. We appreciate the reviewer's suggestions and literature, which we inserted into the introduction and the discussion of the results in section 3.2.
* * ** * *
**Point-by-point responses to Reviewer #1:**

**Specific Comments:**

1. **Line 268- All components of a Rossby wave packet do not necessarily travel together since the Rossby wave is dispersive.**

We meant that the relative contributions of the different wavenumbers of a travelling Rossby wave packet remain more or less similar during its lifetime, since the troughs-ridges do not expand-contract in longitude (e.g. in Hovmoller diagrams of upper-tropospheric extratropical RWPs). We rephrased this sentence into: *'Strictly speaking, the wavenumber composition of a travelling Rossby wave packet should remain similar during its lifetime: the dominant wavenumber does not change. Our analysis shows that this is not the case with W3 which becomes dominant.'*

2. **Line 300 - It seems somewhat contrived to choose the 0.066/day peak to discuss the W3 being locally generated when that is not the frequency with the strongest power. Where does the power at 0.055/day come from?**

    +

3. **Line 303 - I am also somewhat unclear on the following: you claim that a combination of the fast W2 (momentum flux Fig. 10c) and slow W1 (amplitude Fig. 10e) create the slow W3. When you say fast W2 are you talking about the w0520 Rossby wave? The 0.033/day frequency implies a period of 30 days, which is inconsistent with your "fast W2" resonance theory. Can you elaborate?**

Thank you for pointing this out. We agree with your and reviewer #3 comments that the results in Fig. 11 and the corresponding discussion in section 3.1.3 contained some inconsistencies, which we didn't realize previously. We've performed a much more detailed analysis, shown in the new Fig. 11 and we've rewritten the corresponding part in section 3.1.3. It turned out that all wavenumbers have

changing speeds/frequencies during their lifetime in early 2016, therefore a single peak in their Fourier spectrum (Fig. 11 in the previous manuscript) was not necessarily representative of what was happening in February 2016.

In the new Fig. 11 it is shown that, despite the changing wave frequencies, resonance conditions (dark blue in Fig. 11d) were present for a fair amount of time in the subtropics and near the equator during late January and the first half of February, coincidentally with the amplification and the strongest momentum fluxes of W3 (Fig. 10d). In other latitudes and times, resonance conditions (dark blue) do not last more than a couple of days in a row.  We hope that the current analysis of the resonance theory is more convincing now.

4. **Line 350- Enhanced with respect to what? The previous winter? Of course the Kelvin wave activity depends on the wind structure. There are no Kelvin waves here until February because the winds are too strong westerly. The general tropospheric Kelvin wave activity looks to be lower than the previous winter too.**

     +

5. **Line 356 - Again, is the Kelvin wave momentum flux "enhanced" just because Kelvin waves can propagate into this region of weak westerlies? If yes, this is not surprising.**

We rephrased these two paragraphs using 'above-average' instead of 'enhanced' for clarity.

6. **Line 360 - This is not necessarily true. Kelvin waves can propagate through westerly flow as long as their phase speed is greater than the flow speed. So the fact that the westerlies are weak in this case allows them to do so.**

Many thanks for this comment, this makes more sense. We substituted the reasoning for vertical Kelvin wave propagation within weak westerlies accordingly.

7. **Line 374 - In the standard QBO paradigm, increased Kelvin wave activity causes descent of the westerlies, because the waves accelerate the flow below their critical level, which in turn lowers the critical level, and so on. Why in this case does the increased Kelvin wave activity not promote descent?**

We added the following discussion at the end of the paragraph: *'Typically, Kelvin wave momentum deposition promotes downward propagation of the QBO westerly phase, since the waves encounter westerly shear and a rather narrow region below the critical level near the bottom of the QBO*

*westerlies where the Kelvin wave momentum is deposited. This is not the case in early 2016, with weak westerlies (weak easterlies later on), low shear between 100 hPa and 20 hPa, and easterly shear above. These sub-critical conditions make the height range for Kelvin wave momentum deposition quite broader than usual, thereby enabling a prolongation and even upward propagation of the westerly QBO phase. This interpretation would be in agreement with Das and Pan (2015), who found faster-descending QBO westerly phase only when the lower stratospheric winds favored Kelvin wave upward propagation (not the case in early 2016).'*

8. **Line 394 – There are a few studies to have linked Kelvin wave activity to El Nino. Cf. Yang and Hoskins (https://doi.org/10.1175/JAS-D-13-081.1), Das and Pan (https://doi.org/10.1016/j.scitotenv.2015.12.009), Rakhman et al (doi:10.1088/1755-1315/54/1/012035)**

Thank you for this suggestion and for the valuable references. We removed the sentence '*To our knowledge, the enhanced Kelvin wave activity and its relation to ENSO has not been noted previously.'* which was wrong, and we added the references into the previous paragraph discussing Fig. 15.

**Technical Corrections:**

9. **Figure 1 caption - Please delete the first instance of "The vertical blue lines denote the time periods..." in the first sentence (it is repeated later). In panel (a) there are no red triangles, so either add them or do not mention them in the penultimate sentence.**

Corrected.

10. **Figure 5 - The label should read "Total Rossby wave" (typo)**

Corrected.

11. **Figure 8 - Can you add a legend to this figure? Also, are the time axes correct? You mention on Line 241 different months for the peaks in each year, but these all peak in February.**

Corrected. Note that in the sentence referring to Fig. 8, the peaks where meridional propagation is most clear are mentioned, not necessarily the strongest ones. We rephrased this sentence for better clarity.

12. **Figure 10 caption - Please move the sentence about Panel (e) to the end, the descriptions that follow it all refer to the other panels so it is a little confusing.**

Corrected.

13. **Figure 11 caption - "February" (typo)**

Corrected.
* * ** * *
**Point-by-point responses to Reviewer #2:**

**General comments: First of all, this manuscript can benefit from some revisions and editing to improve for clarity and conciseness. Some recommendations are provided in the comments below. As everyone's writing style is different, those comments are by no means requirement but rather indications where writing could benefit from some editing to make your massage clearer. Second, some clarifications should be done in Data and Methods section. Line by line comments are provided below. For example, I strongly recommend to rewrite "Wave filtering" subsection in a more clear way as it can be very confusing to some. In several places, your statements like "It is difficult to (do something) " can be confusing as it is not very clear if you've applied your technique to the dataset. How did you do your filtering or any other analysis step by step? State these steps first without any unnecessary information and then provide any relevant comments regarding these techniques at the end. See more detailed comments below. Furthermore, Introduction or/and Discussions are lacking a discussion of literature on relationship between ENSO and Kelvin waves. While a role of enhanced Kelvin Wave activity due to strong 2015/2016 El Nino event in producing the extended westerly zonal wind near 20 hPa is definitely new (and very interesting) result, the enhanced Kelvin wave activity and its relation to ENSO definitely was previously looked at and documented. For instance paper by Yang and Hoskins (2013) along with others should be mentioned and cited. [Yang, G. and B. Hoskins, 2013: ENSO Impact on Kelvin Waves and Associated Tropical Convection. J. Atmos. Sci., 70, 3513–3532, https://doi.org/10.1175/JAS-D-13-081.1]**

Many thanks for your suggestions and new literature. We've addressed all the detailed line-by-line comments below.

**Specific comments:**

1. **Line 95: Why is combined ERA-40 and ERA-Interim reanalyses are used instead of just using ERA-40 or ERA-Interim reanalyses by itself. It would be useful for readers to see a short statement explaining benefits and limitations of individual datasets vs combined one.**

   The sentence now reads: *'For this study, we use the combined European Center for Medium-Range Weather Forecasts (ECMWF) ERA-40 and ERA-Interim reanalysis data sets (Uppala et al. 2005, Dee et al. 2011), which extends from 1958 to 2017. This was done to use the longest timeseries up to the 2016 QBO reversal to cover as many QBO cycles and ENSO events as possible, instead of starting from 1979 with ERA-Interim alone.'*

2. **Line 96: Using layer averaging (30-50 hPa) to represent 40hPa level is fine if 40 hPa level is not available as an output. However, this (or other reasons for layer averaging) should be mentioned.**

   We mention this now in the sentence.

3. **Line 106: "similar to U40". Is U20 also defined as a layer averaged U between two pressure levels? Please clarify.**

   We added *'Both ERA-40 and ERA-Interim provide the 20 hPa level so no averaging is needed'* for clarity.

4. **Line 115-117: "Previous studies... " How this sentence connects to the filtering techniques you are discussing? This seems an odd place for comparison with previous studies unless you are comparing techniques that you are using in this study with one's in previous studies.**

   We agree with the reviewer and this sentence has been deleted.

5. **Line 118: "following dispersion curve" Is it a part of the "kf-filter" function technique or something you performed after applying this filter. If it is unrelated to the "kf-filter" than Line 112 shouldn't be a fist sentence of this paragraph.**

   +

6. **Line 118, also line 124 A use of statements like "it is difficult … " or "we can .." are very confusing and distracting since I am not sure if you applied the technique to your data or just speculating about it. Make sure you are clearly explaining your methods.**

We've rewritten parts of section 2.2 to improve clarity, especially in the first half. We hope the filtering method is easier to follow now.

7. **Line 173: "was weaker". What do you mean by this (its amplitude or persistence in time)? To my eye these three events don't stand out very well when compared to other years in this record but I could be missing something. Being specific may help.**
We rephrased this sentence as follows:
*'Figure 1b shows that the amplitude of westerly zonal wind was weaker in a number years (e.g., 1959/1960, 2010/2011) but the westerly zonal wind only reversed its direction in the 2015/2016 at 40 hPa.'*
There are many other events where westerlies are weaker than usual, as the reviewer is pointing out, but the ones we picked also show strong deviations of Rossby wave momentum fluxes (Fig. 1d). This is shortly mentioned in the next paragraph but now we highlight this feature earlier on.

8. **Line 184: "However, we cannot ..." very wordy sentence. How about something like this : "Since enhanced Rossby wave behavior doesn't extend to the lower and higher levels, the observed signal less likely originated from vertical propagation." Side note: Please comment a strong signal in enhanced Rossby wave activity during October 2015.**
Corrected.

Section 3.1 focuses on motivating the subsections 3.1.1, 3.1.2 and 3.1.3 and the analyses made on the 2016 QBO reversal, we feel that commenting on the Rossby $U^2$ anomaly increase of October 2015 will only distract the reader.

Later in Figures 3 or 6, one can track the momentum flux anomalies of October 2015 to wave activity originating in the SH extratropics (blue colors). However, the Oct. 2015 momentum flux anomalies are not as strong in magnitude as the ones from February 2016: note they are not necessarily proportional to $U^2$ anomalies from Fig. 2. In addition, the anomalous fluxes in Oct. 2015 do not seem to disrupt the westerly QBO core in the tropics, and thus they are not of high relevance for our study.

9. **Line 224: What do you mean under "it's different components". What components exactly do you mean? Be more specific.**

   We meant the Rossby waves with different wavenumbers (1, 2 and 3). We rephrased the sentence as follows:

   *'... and our analysis will describe the evolution of the Rossby waves with wavenumber 1, 2 and 3, independently.'*

10. **Line 236: What do you mean by "weaker but similar"? These are two contradictive words. Weaker in strength (or amplitude) but similar in shape?**

    We meant wavenumbers 1 and 2 had a similar contribution, but anyways weaker than the other cases, but we see that this was confusing. We removed 'but similar' from the sentence.

11. **Line 245: "slower than 0.15 cpd, or 6-7 day frequency". Should it be day-1 ? Also, can you change these to a more familiar timescales which you are using (in days)? This will make comparison with your study easier and you'll support your augment for their study to include both, the faster and quasi-stationary Rossby waves". Otherwise it is not very obvious.**

    We apologize for the mix-up, we were referring to 6-7 day periods of course. Now the sentence says: *'(with periods above 6 days)'*, and we also corrected freq.--> perdiods at the end of the paragraph.

12. **Line 325: Very confusing. Please rewrite. First sentence of the paragraph should reflect its main idea or statement.**

    The sentence *"The reversal event not only depends on the enhanced Rossby wave activity in the tropics but also on the different background zonal winds."* is moved to the start of the paragraph.

13. **Line 351: "50 to 20 hPa" Do you mean "50 -200 hPa" because I see a strong signal in the upper troposphere as well?**

    We've rewritten the entire sentence for better clarity:

    *'Figure 13 shows a predominance of above-average Kelvin wave squared perturbations in zonal wind (mostly red and yellow shading and absence of blue) in the upper troposphere around 200 hPa and 100 hPa throughout 2015 and early 2016. The upper tropospheric Kelvin waves can be seen propagating upward within easterly winds most of the time. Also, from December 2015 to April 2016, vertical propagation of Kelvin waves can be observed within weak westerly winds*

*between 50 hPa and 20 hPa, which will be discussed in more detail below.'*

13.     **Line 384 and also 393-394: As mentioned in General Comments, this is definitely not a new result. Previous studies, discussing enhanced KW activity during El Niño events, have to be cited and also discussed. One of these studies is by Yang and Hoskins (2013). Yang, G. and B. Hoskins, 2013: ENSO Impact on Kelvin Waves and Associated Tropical Convection. J. Atmos. Sci., 70, 3513–3532, https://doi.org/10.1175/JAS-D-13-081.1**

Many thanks for this reference, it's been added among others into this section. Particularly the last paragraph of section 3.2 has been rewritten discussing this topic.
* * *
**Technical comments:**

1. **Line 136: "In order to explore" -> "To explore" (remove "In order")**
   Corrected.

2. **Line 152: "We explore a possible ...... in section 3.1.3". -> remove this sentence**
   Sentence removed.

3. **Line 180: "Figure 2 illustrates .... ." Remove entire sentence since the same information is included in Figure 2 caption.**
   Sentence removed.

4. **Line 183: "Enhanced Rossby wave ... in the NH" -> remove the entire sentence (it doesn't have a logical connection with sentences before and after)**
   Corrected.

5. **Line 202: "Figure 4 shows that the horizontal .." -> "As expected, the horizontal Rossby wave momentum fluxes peak in the ..."**
   Corrected.

6. **Line 211: "We calculated the mean ...". -> "The mean horizontal ... in the tropics are shown in Figure 5. The maximum ..."**
   Corrected.

7. **Line 223-224 "Their study was ..." -> "Although their study ... (35.8 hPa and 40 hPa respectively), we describe the evolution ..."**

   We kept this part of the sentence as is, as we want to emphasize the similarity of our study with Lin et al. 2019.

8. **Line 229-230: Remove "As shown in Figure 5" and connect it with next sentence -> "The Rossby waves ... (red triangles in Figure 5) with horizontal momentum flux having maximum values of about ... ."**

   Corrected.

9. **Line 233: "We calculated ... " -> "To explore the sources of quasi-stationary Rossby waves, Figures 6 and 7 show the horizontal ...."**

   Corrected.

10. **Line 234: remove entire sentence "Then we analyzed ..."**

    Corrected.

11. **Line 235-237: Remove "Figures 6 and 7 show" and combine this sentence with a next one -> "The contribution of Rossby waves .... in November 2010, while ... "**

    Corrected.

12. **Line 239: Put a dot after "in Figure 8", remove "which reveals that"**

    Corrected.

13. **Line 240-242: change to : "The peaks of Rossby waves w2040 in Jan 1960, Nov 2010 and Feb 2016 at 40 hPa occurred earlier at higher latitudes, which is indicative of their extratropical origins. This is in agreement with Figures 5, 6, and 7 revealing an important contribution of quasi-stationary Rossby waves to the enhanced Rossby activity in the tropics mainly by wavenumber 1 and to a lesser degree by wavenumber 2 of extratropical origin."**

    We rephrased this sentence slightly differently making clearer that we refer to the peaks whose equatorward propagation is best visible, not all the strongest ones (to also account for comment 11 by reviewer #1).

14. **Line 252-260: Need some revisions for conciseness. For example, there are few sentences that can be combined together into one and extra words have to be removed (e.g., "as shown in" (line 256), "the results reveal that in particular" and so on).**

We rephrased the sentences as bellow in the manuscript:

Figure 9 shows that the amplitude of the horizontal Rossby waves w0520 momentum flux was stronger in extratropics in January 1960 and November 2010, while there were two peaks in February 2016 which is in extratropics (around 40°N) and in the tropics (around 15°N). We therefore focus now on the contributions of W1, W2 and W3 separately during the 2016 reversal event by analyzing the time-latitude cross section of their horizontal momentum flux from November 2015to April 2016 at 40 hPa (Figure 10). The result reveals that the activity of W2 and W3 was enhanced while W1 was very weak in early February 2016. The maximum horizontal momentum flux of W2 occurred around 40°N in early February 2016 (Figure 10), and propagated equatorward with some days lag between 40N and 15°N (see Figure S4), indicating its extratropical origin and equatorward propagation.

15. **Line 284: How about : "The meridional gradients are negative in late March 2016, which indicates barotropic – baroclinic instability (see section 2.4). In this study, W3 maximizes one month prior to March 2016. This is consistent with Coy et al. (2017)**

Corrected slightly differently to also account for reviewer #3 technical comments.

16. **Line 341: Start this sentence with "The average period ..." and remove all words before it ("By performing ... we found that").**

Corrected.

17. **Line 342: Add -> "(Figure 12). However, in 2015/2016 the ..."**

Corrected.

18. **Line 351: Remove "Figure 13 shows that ...pertubations had" -> "from 50 to 200 hPa with maximum amplitudes ..."**

This paragraph has been rephrased for better clarity.

19. **Line 380: remove parenthesis around 20 hPa**

Corrected.

20. **Figures: It would be great if you can increase thickness of the zonal mean zonal wind contours.**

Corrected.
* * ** * *
**Point-by-point responses to Reviewer #3:**

**Major comments**

1. **Usually, to capture the propagation and interaction of global scale waves Eliassen Palm (EP) fluxes are calculated. You are calculating only  (vertical momentum flux) and  (horizontal momentum flux) which are only part of the EP flux and do not mention the limitations of this approach.**

   We've rewritten a big part of section 2.3 to address this: now we state our method's limitations but at the same time we also reassure the reader that for the purposes of our study it is sufficient. Related to this, we also appreciate the reviewer's suggestions in specific comment #7, which we've introduced into section 2.3.

2. **The whole study is based on the kf-filter method. However, almost no information is given how this method was applied.**

   Parts of section 2.2 have been rewritten to increase clarity about the options used with the kf-filter, we hope it is easier to follow now.

**Specific Comments:**

1. **General comment: When discussing momentum fluxes, you often do not state clearly what you are discussing - the momentum fluxes, or their anomaly. This happens in the text, as well as in the figure captions and in the figure legends! In my comments, I mentioned only a few occurrences because they are too many. Please revise carefully throughout!**

   We apologize for the unclear presentation of the momentum flux anomalies, we've corrected the terminology throughout the manuscript.

2. **General comment: 20hPa (about 27.5 km altitude) is in the middle stratosphere, not the upper stratosphere. The stratosphere extends from about 10-20km to about 50km. Please revise carefully throughout the paper!**

Corrected throughout the manuscript.

3. **p.2, l.44-46 here you write: The eastward propagating Kelvin waves provide the main eastward acceleration for the initiation of the QBO westerly phase. In contrast, the westward propagating Rossby waves provide the main westward acceleration for the initiation of the QBO easterly phase.**
**This is not entirely correct because also small scale gravity waves contribute. Mainly both Kelvin waves and small scale gravity waves contribute to the forcing of the QBO westerly phase, while global scale westward traveling tropical waves and small scale gravity waves contribute to the forcing of the QBO easterly phase (see for example Ern and Preusse, GRL, 2009 and Ern et al., JGR, 2014, Garcia and Richter, JAS, 2019).**
**References:**

**Ern, M., and P. Preusse, Quantification of the contribution of equatorial Kelvin waves to the QBO wind reversal in the stratosphere, Geophys. Res. Lett., 36, L21801, doi:10.1029/2009GL040493, 2009.**

**Ern, M., F. Ploeger, P. Preusse, J. C. Gille, L. J. Gray, S. Kalisch, M. G. Mlynczak, J. M. Russell III, and M. Riese, Interaction of gravity waves with the QBO: A satellite perspective, J. Geophys. Res. Atmos., 119, 2329-2355, doi:10.1002/2013JD020731, 2014.**

**Garcia, R. R., and J. H. Richter, On the Momentum Budget of the Quasi-Biennial Oscillation in the Whole Atmosphere Community Climate Model, J. Atmos. Sci., 76, 69-87, 2019.**

Many thanks for the references and additional detail for this paragraph. The sentences were rephrased in the introduction as follows:

*'The eastward propagating Kelvin waves and small-scale gravity waves provide the main eastward acceleration for the initiation of the QBO westerly phase, while the westward propagating Rossby waves and small-scale gravity waves provide the main westward acceleration for the initiation of the QBO easterly phase (Ern et al., 2009, 2014; Garcia et al., 2019).'*

4. **p.3, l.68: These references address only global scale waves. A reference for smallscale gravity waves should be added. Using model simulations, differences in the QBO forcing by gravity waves for different ENSO conditions were investigated, for example, by Kang et al., JAS, 2018.**
**Reference:**
**Kang, M.-J., H.-Y. Chun, Y.-H. Kim, P. Preusse, and M. Ern, Momentum Flux of Convective Gravity Waves Derived from an Offline Gravity Wave Parameterization. Part II: Impacts on the Quasi-Biennial Oscillation, J. Atmos. Sci., 75, 3753-3775, 2018.**
Thank you for the reference, it has been added.

5. **p.3, l.91: Please state more clearly that a merged ERA-40/ERA-Interim dataset is used to have a longer time series because ERA-Interim starts with the year 1979 and does not cover the earlier period.**
The sentence now reads: *'For this study, we use the combined European Center for Medium-Range Weather Forecasts (ECMWF) ERA-40 and ERA-Interim reanalysis data sets (Uppala et al. 2005, Dee et al. 2011), which extends from 1958 to 2017. This was done to use the longest timeseries up to the 2016 QBO reversal to cover as many QBO cycles and ENSO events as possible, instead of starting from 1979 with ERA-Interim alone.'*

6. **p.4, l.114/115: You should elaborate much more on the kf-filter! As far as I understand, for a fixed latitude and altitude, you enter the whole time series of 60 years with a longitude resolution of 2.5 deg into the kf-filter. The time series is tapered to zero at both ends. Which function is used? Split-cosine-bell as stated in https://www.ncl.ucar.edu/Document/Functions/Built-in/taper.shtml? Did you use the standard settings of the taper of p=0.1? In this case, you would have to discard 3 years at each end of the time series and valid data are obtained only for 1961-2014. This would contradict your statement on p.5, l.139 that only the years 1958 and 2017 would be affected. Please explain!**
The taper's p parameter cannot be chosen when using the kf-filter function (see *https://www.ncl.ucar.edu/Document/Functions/User_contributed/kf_filter.shtml* ), there's no input argument for the taper.
We did check the different filter outputs and cutting 1 year of data seemed perfectly fine: see the two graphics below for Kelvin u2 at 200hPa (10S-10N) and Rossby w0570 at 40hPa (35-45N)

Also, parts of section 2.2 have been rephrased to improve clarity, but we feel that details such as the taper settings belong in the kf-filter / taper documentation to which the readers are referred to.

[Figure]

**Figure. The temporal variation of squared Kelvin wave zonal wind perturbations at 200 hPa. The vertical dashed lines denote the January of 1959 and 2017.**

[Figure]

The timeseries of Rossby wave (with 5-70 day periods ) squared
meridional wind perturbations at 40hPa averaged for 35-45°N

RPK: I went through the code of the kf-filter ncl function ( *https://k3.cicsnc.org/carl/carl-ncl-library/blob/24eb0d1bf7913c8d63fb7d43ad5e2b298269b931/kf/kf_filter.ncl* ) and in line 49 the tapering is set:

tempData = taper( tempData, 0.05, 0 )  → Option '0' is for tapering to the series mean instead of to zero. This is inside the kf_filter.ncl function, and not for the user to choose.

With p=0.05 only 1.5 years of the data would be affected, and taking into account the cosine-bell shape, the data 1-1.5 years away from the timeseries ends is affected relatively much less. In any case, the first half of 2016 is not affected by the taper.

As mentioned above, and since this issue does not affect our results, we believe the inclusion of such intricate details about the kf-filter is unnecessary.

7. **p.5, about section 2.3: Please explain why you calculate momentum fluxes in this way! Possibly, you are missing something or biasing your analysis by not calculating the full EP flux vector! Therefore at least the limitations of your approach should be clearly stated. Usually, for global scale waves EP fluxes are calculated to capture their propagation and effect on the background flow, however you are neglecting heat fluxes. By calculating  you are using the quasi-geostrophic approximation of the EP flux in meridional direction (see for example Matthias and Ern, ACP, 2018). This approximation can be used for extratropical Rossby waves and should be sufficient for diagnosing the meridional propagation direction of Rossby waves from the extratropics. In the tropics, however, wave motions can become ageostrophic, and this approximation may no longer hold. This is the case for tropical waves like Kelvin waves or tropical Rossby waves.**
   **The term  is part of the vertical component of the EP flux and should be a good approximation for Kelvin waves because v' is zero for these waves.**
   **Reference: Matthias, V., and M. Ern, On the origin of the mesospheric quasi-stationary planetary waves in the unusual Arctic winter 2015/2016, Atmos. Chem. Phys., 18, 4803-4815, 2018.**

   Many thanks for the suggested reference, now added into section 2.3, and for pointing out these issues with the momentum flux calculations. We completely agree with the reviewer about the limitations of our method, and we've rewritten section 2.3 to state that this is a simplified approach, nevertheless sufficient for the goals of our study. We also mention that, despite ageostrophy holding near the equator, we see no discontinuities in the time-latitude sections of Rossby wave activity crossing into the SH (e.g. Figs. 3 and 10).

8. **p.5, l.149/150: here you state that Rossby waves would not contribute much to  as could be seen from Figs. S1 and S2 in the supplement. In the supplement, however, only "anomalies" of  and  from their monthly means are shown.**
        +

9. **captions of Figs. 3, S1 and S2: First you write that horizontal momentum fluxes would be shown, and later in the caption you write that color shadings would be anomalies with respect to the monthly climatology. This is very confusing, please write more clearly!**

   We apologize for the inconsistencies. We've changed Figs. S1 and S2 into momentum fluxes (not anomalies) and we've updated figure captions and sentences throughout the manuscript stating that what's shown are anomalies from climatology. We hope it's not confusing any more in the new manuscript version.

10. **Follow-up question: Were the monthly climatologies temporally interpolated to single days to avoid jumps in the anomalies from one month to the other? Please clarify!**

We kept the monthly climatology without interpolation, as we didn't see any jumps of importance in any figure.

11. **p.5, l.152/153 and l.155: This is not entirely correct: Shuckburgh et al. (2001) investigated only barotropic instabilities, not baroclinic instabilities! Only Coy et al. (2017) included also baroclinic instabilities.**

We rephrased this paragraph as follows:

*'Shuckburgh et al. (2001) revealed that barotropic instability in tropical regions could be associated with QBO westerlies. Coy et al. (2017) later studied both barotropic and baroclinic instability following Andrews et al. (1987). We use a similar approach calculating the meridional gradient of potential vorticity in 2015/2016  from 70 hPa to 10 hPa following Andrews et al. (1987) and Coy et al. (2017).'*

12. **p.7, l.210: Why did you select w2040 for the quasi-stationary Rossby waves, and not w2070? Choosing w2070 would be much more intuitive!**

                +

13. **p.7, l.214-219: Would also the contributions of w2070 be similar in all three cases? Please note that w0520 and w2040 do NOT sum up to the "total Rossby" fluxes! Possibly w4070 might show also considerable case to case variability.**

We originally divided the total Rossby wave into three kinds: w0520, w2040 and w4070.  We found that Rossby w4070 had near-zero values in all cases, therefore we didn't include it in the figures. We now mention this more clearly in sections 2.2 and 3.1 in the manuscript. We repeated Fig. 5 from the manuscript including w4070 (see below), but we'd advocate to keep the less busy one (without w4070) for the main manuscript.

[Figure]

Please also note that ($U_a V_a + U_b V_b$) is not necessarily equal to ($U_a + U_b)*(V_a + V_b$), this is why the Total Rossby wave momentum flux is not the linear sum of the momentum fluxes from w0520+w2040.

14. **p.10, l.305: here you state: "the quasi-stationary W1 and the faster W2 which came from the extratropics, generated W3 locally..." This contradicts your statements from above that W3 was generated by W1 and W2 of 0.033 day^-1 which are both in your quasi-stationary range of frequencies. Please clarify!**

We performed a more detailed analysis in the new Fig. 11 and rewritten the interpretation of the wave resonance mechanism in section 3.1.3. It turned out all W1-3 are changing speeds/frequencies throughout January-March 2016 and that a single Fourier spectrum did not entirely represent what was happening during February 2016. However, the main conclusion remains similar.

15. **p.10, l.313/314: here you state that: "W3 cannot be an equatorial wave mode in principle, otherwise its amplitude would be by definition maximizing at the equator." I think that this statement does not generally hold. Equatorial Rossby modes can be symmetric or anti-symmetric with respect to the equator. Therefore either u' or v' could be zero at the equator, leading to zero . For a survey of equatorial modes see, for example, Yang et al., JAS, 2003, their Fig.3.**
**Reference: Yang, G.-Y., B. Hoskins, and J. Slingo, Convectively Coupled Equatorial Waves: A New Methodology for Identifying Wave Structures in Observational Data, J. Atmos. Sci., 60, 1637-1654, 2003.**

We agree with the reviewer and we deleted the sentence in the manuscript.

16. **p.10, l.325: Enhanced momentum flux does not necessarily mean that the zonal wind is accelerated or decelerated. There is only an effect on the background flow when this momentum is deposited (if there is a non-zero divergence of the EP-flux). Of course, enhanced momentum fluxes can lead to stronger EP-flux divergences.**

Also related to specific comment #7: we state this now within section 2.3 as another limitation of our methods.

17. **About Figs.13 and 14: As Kelvin waves in the stratosphere are modulated rather by the QBO than by a seasonal cycle, does it really make sense to show deviations of Kelvin wave amplitudes or momentum flux from a monthly mean climatology?**

In this case it is useful for commenting on upper-tropospheric anomalies, which do show a seasonal cycle, and to discuss their relation to El-Nino as done later in Fig. 15.

18. **p.12, l.360/361: Another possibility could be that these Kelvin waves have phase speeds that exceed the westerly wind. As can be seen, for example, in Ern et al., ACP, 2008, Kelvin waves with high phase speeds are not much modulated by the QBO, while slower phase speed Kelvin waves are strongly modulated by the QBO with minimum amplitudes during westerly winds.**
**Reference: Ern, M., P. Preusse, M. Krebsbach, M. G. Mlynczak, and J. M. Russell III, Equatorial wave analysis from SABER and ECMWF temperatures, Atmos. Chem. Phys., 8, 845-869, 2008.**

Many thanks for pointing this out and the reference, of course this makes more sense. We added the following explanation in section 3.2:
*'Also, if the Kelvin wave's phase speed exceeds that of the westerly flow, it is possible that the*

*Kelvin wave propagates upward within weak westerlies as reported by Ern et al. (2008). '*

19. **p.13, l.417/418: This is not correct: Both W1 and W2 have the same frequency and fall in the frequency range that you call quasi-stationary.**

    We refined the analysis in our new Fig. 11 and rewritten the corresponding discussion in section 3.1.3, which now do support this conclusion.

##

**Other Comments:**

1. **p.2, l.35: The westerly mean flow in the tropical stratosphere generally favors -> If the mean flow in the tropical stratosphere is westerly, it generally favors**

    Corrected.

2. **p.5, l.150: Fig.12a does not exist! Should this read Fig. S1? Please check!**

    Thank you for your remark, it should have been Figure 14a. We have corrected this in the manuscript.

3. **Caption of Fig.1, l.2: ??? horizontal Rossby wave momentum flux -> Rossby wave horizontal momentum flux anomaly**

    Corrected.

4. **Caption of Fig.1: Different from what is stated in the caption, there are no red triangles in Fig.1a**

    Corrected.

5. **p.6, l.180 / Fig.2: It is unclear what is shown! Probably your notation is misleading! In the figure legend of Fig.2 it reads: <du'ˆ2>, suggesting that you calculate a climatological u' distribution, and you are showing deviations from that average u' for a particular period. In this case, however, there could be no negative values because values are squared for display! So my guess is that it should read d in the figure legend. Similar, in Fig.3 it should probably read d instead of <du'v'>.**

    Yes, as you said Figure 2 shows the deviations of the squared zonal wind perturbation respect to the climatology. We rephrased the sentence as follows:
    *'Figure 2 illustrates the time-height cross section of the deviations of the squared Rossby wave*

*anomalies in zonal wind respect to the monthly climatology (1959-2014) averaged over tropics (10°N-10°S).'*

We've also applied the correct terminology **d** in all corresponding figures, we apologize for the unclear presentation in the previous manuscript version.

6. **p.6, l.190: horizontal Rossby waves momentum flux -> Rossby wave horizontal momentum flux anomalies**

   Corrected.

7. **Fig.5: Please mention in the figure caption that "total Rossby wave" corresponds to w0570.**

   Corrected.

8. **p.8, l.251: momentum flux -> momentum flux anomaly**

   Corrected.

9. **p.8, l.251, suggested rewording, as this cannot be seen from Fig.9: The contribution of Rossby waves w0520 is largest in February -> In the tropics, the largest anomaly of Rossby waves w0520 is found in February**

   Corrected.

10. **p.8, l.252: momentum flux was stronger -> momentum flux anomaly was stronger**

    Corrected.

11. **caption of Fig.10, l.1: horizontal momentum fluxes -> horizontal momentum flux anomalies**

    Corrected.

12. **Caption of Figs. S5 and S6: Please mention the altitude/pressure of these sections.**

    Corrected.

13. **p.9, l.283: and below -> and at higher altitudes**

    Corrected for 'at lower altitudes' since we refer to levels below 40 hPa throughout this paragraph (at levels 20 hPa and above, there are little structures to comment on and q meridional gradient is mostly positive).

14. **p.9, l.284: period below 40 hPa. -> period at pressures below 40 hPa.**

    Corrected.

15. **p.9, l.284: The meridional gradient starts -> At 40 hPa the meridional gradient starts**

    Corrected.

16. **p.9, l.285: barotropic and baroclinic -> barotropic and/or baroclinic**

    Corrected.

17. **p.10, l.298: W3 has the most complex peaks, the stronger peaks are corresponding to the frequencies are at -> W3 has a broad peak with the strongest contributions at**

    This paragraph has been completely rewritten due to the inclusion of the new figure 11.

18. **In Fig.14b: why are the lines interrupted? Are no-shows insignificant values? If yes, please state in the figure caption!**

    We only show the westerly zonal wind condition. The easterly wind and its corresponding Kelvin wave momentum fluxes are not shown. We rephrased the figure caption to make this clearer.

---

## Author Response (AR2)

**Response to reviews of "On the forcings of the unusual QBO structure in February 2016" by Haiyan Li et al.**

Dear Editor,

We would like to thank the three anonymous reviewers for their comments and suggestions, which led to a substantial improvement of the manuscript's quality. In the following paragraphs, we include our point-by-point responses (in dark gray) to **reviewer #2 comment (bold black).**

Your sincerely,

Haiyan Li

Robin Pilch Kedzierski

Katja Matthes

**Point-by-point responses to Reviewer #2:**

1. **p3, l.21 & p 15, l.25: Center->Centre**
   Corrected.
2. **p.4, l.3: (following Hansen et al. (2016). -> (following Hansen et al. (2016)).**
   Corrected.
3. **p.4, l.28ff: here you write: "where the propagation of waves is highly affected by the background winds through Doppler shifting, and thus a single Rossby wave mode can occupy any region in the wavenumber-frequency domain depending on the background wind condition."**
   **To my knowledge, Doppler shifting by the background wind will mainly change the intrinsic frequency, and not so much the ground based frequency of a wave. Therefore I think the above statement is too strong. #### --> change to 'can occupy different regions in the wavenumber-frequency domain...'**
   Corrected.
4. **p.11, l.14 there's a -> there is a**
   Corrected.
5. **Units are missing in Figs. S4 - S7**
   Corrected.